# Keyhole fluctuation and pore formation mechanisms during laser powder bed fusion additive manufacturing

Yuze Huang [1,2✉], Tristan G. Fleming [3], Samuel J. Clark [1,2,4], Sebastian Marussi[1,2], Kamel Fezzaa[4], Jeyan Thiyagalingam [5], Chu Lun Alex Leung [1,2✉] & Peter D. Lee [1,2✉]

Keyhole porosity is a key concern in laser powder-bed fusion (LPBF), potentially impacting component fatigue life. However, some keyhole porosity formation mechanisms, e.g., keyhole fluctuation, collapse and bubble growth and shrinkage, remain unclear. Using synchrotron X-ray imaging we reveal keyhole and bubble behaviour, quantifying their formation dynamics. The findings support the hypotheses that: (i) keyhole porosity can initiate not only in unstable, but also in the transition keyhole regimes created by high laser power-velocity conditions, causing fast radial keyhole fluctuations (2.5–10 kHz); (ii) transition regime collapse tends to occur part way up the rear-wall; and (iii) immediately after keyhole collapse, bubbles undergo rapid growth due to pressure equilibration, then shrink due to metal-vapour condensation. Concurrent with condensation, hydrogen diffusion into the bubble slows the shrinkage and stabilises the bubble size. The keyhole fluctuation and bubble evolution mechanisms revealed here may guide the development of control systems for minimising porosity.

[1] UCL Mechanical Engineering, University College London, WC1E 7JE London, UK. [2] Research Complex at Harwell, Harwell Campus, Didcot OX11 0FA, UK. [3] Department of Physics, Queen's University, Kingston, ON K7L 3N6, Canada. [4] X-ray Science Division, Argonne National Laboratory, Lemont, IL 60439, USA. [5] Science and Technology Facilities Council, Harwell Campus, Didcot OX11 0FA, UK. ✉email: yuze.huang@ucl.ac.uk; alex.leung@ucl.ac.uk; peter.lee@ucl.ac.uk

Laser powder-bed fusion (LPBF) additive manufacturing is being widely explored in both industry and academia[1,2] for the production of metal parts. During LPBF, mid-power (~ 100–1000 W) but tightly focused (spot sizes ~ 20–100 μm) lasers are scanned across successive layers of fine metal powder at high speed (~0.05–4 m s$^{-1}$), selectively melting and consolidating the powder to build a fully dense part. The typical processing-structure–property linkage for LPBF is: steep thermal gradients and high cooling rates[3] (~10$^4$–10$^6$ K s$^{-1}$) favouring fine, columnar grains oriented along the build direction, producing as-printed LPBF parts that typically exhibit increased strength, reduced ductility, and increased microstructural and mechanical property anisotropy[4], depending on the alloy systems.

The laser fluence during LPBF is sufficient to vaporise the metal, generating a recoil pressure that pushes molten metal away from the laser–matter interaction zone[5]. With increasing laser fluence, the recoil pressure is large enough to open a deep, high aspect ratio vapour depression, referred to as a keyhole[6]. This is commonly used in laser welding to achieve thin and deep joints[7]. LPBF often operates in keyhole mode melting[6] to ensure complete fusion between successive layers. Additionally, laser absorptivity increases dramatically in keyhole melting due to multiple reflections of the laser beam along the keyhole[8], opening the door for fabrication of highly reflective materials (e.g., aluminium matrix composites with ~91% reflectivity[9]) by LPBF, or enable a more economical laser heat source (e.g., diode laser) to be used in LPBF without sacrificing build efficiency[10]. However, the keyhole is subjected to axial fluctuations and radial perturbations[11] that are governed by the balance of energy and pressure[12–14], posing a significant risk for keyhole instability[15,16] and in some cases, collapse. Keyhole collapse often results in the formation of a bubble in the melt pool, which may get trapped by the solidification front to form a pore. Keyhole pores remaining in the final part may act as stress concentrators and sites for crack initiation and growth, making them potentially detrimental to fatigue life[17] and other final component mechanical properties[18,19].

Several process models[5,12–14,20] explained the physics of keyhole pore formation during laser welding and LPBF, revealing the interactive effects of recoil pressure, surface tension, and Marangoni convection on the keyhole, and the competing influence of gravity, drag, buoyancy and thermocapillary forces on bubble motion. Recently, in situ synchrotron X-ray imaging[21–24] has been applied to LPBF, capturing some dynamics of the keyhole and keyhole pore in the sub-surface of melt pool, including: keyhole morphology evolution[25]; pore formation at turn-around points during raster scanning[26]; pore elimination by thermocapillary forces[27]; pore migration under Marangoni-driven flow and pore coalescence[28,29]; pores being pushed away from the keyhole tip by acoustic waves emanating from a keyhole collapse[16], and pore evolution during multi-layer LPBF[30,31]. However, the dynamics of keyhole pore formation are still not fully understood. The role of keyhole fluctuations in keyhole collapse and the evolutions of bubbles (e.g., formation, growth, shrinkage and migration) before being captured by the solidification front, are largely unexplored. For the latter, previous studies[32,33] explored the influence of evaporation and condensation on the dynamics of water–vapour bubbles in a superheated liquid, and effect of dissolved gas diffusion on bubble growth in casting[34,35], but it remains unclear how evaporation, vapour condensation, and dissolved gas diffusion affect bubble evolution in LPBF.

Here, we perform in situ synchrotron X-ray imaging during LPBF of a commercial aluminium alloy Al7A77 (HRL laboratory, USA), which has critical applications in aerospace, biomedical and automotive industries[19], and also a high laser reflectivity[36] in the near-infra-red, presenting challenges for laser processing. We discover a transition regime (II) between the stable (I) and unstable (III) keyhole regimes in LPBF, where the keyhole morphology changes from wide and shallow in II to narrow and deep in III. Pores are also observed to form in II, mostly present at the rear keyhole wall (RKW), while keyhole porosity is more prevalent in III with pores typically forming at the keyhole bottom. Although some prior work has suggested keyhole fluctuation is largely random, we observe regular oscillations in keyhole width and depth with significant trends in fluctuation frequency across the three keyhole regimes. We find these regimes are well defined by the front keyhole wall (FKW) angle, which collapses to a single function of the normalised enthalpy product[37] for different materials. By comparing our bubble model with experimental data, we find that the bubble dynamics are defined by fast initial growth induced by pressure equalisation, followed by shrinkage due to metal-vapour condensation. Concurrent with condensation, hydrogen may diffuse into the bubble, slowing bubble shrinkage and stabilising the bubble size. Lastly, we investigate the rapid distortion of bubbles as they interact with the advancing solidification front.

## Results

**Keyhole collapse mechanism and related regime transitions.** In situ and operando X-ray imaging was used to probe the keyhole collapse behaviour and keyhole pore formation mechanisms during LPBF, which was carried out using an in situ and operando process replicator (ISOPR, Supplementary Fig. 1), as described in the "Methods" section. We systematically characterised the changes in keyhole shape and bubble development across a wide range of area energy densities AED[38], AED = $P_l/(v_l d_l)$ ($P_l$ laser power, $v_l$ laser scan velocity, $d_l$ laser spot size), from AED = 6 to 17 mJ m$^{-2}$ in the keyhole melting regime[6]. We observed that the keyholes change in morphology from wide and shallow to narrow and deep (Fig. 1a, Supplementary Fig. 2a and Supplementary Movies 1–12). Simultaneously, bubbles first form at the RKW, then prevail at the bottom of keyhole once the keyhole becomes deep and narrow (Fig. 1a, Supplementary Fig. 2a). Those findings indicate that the transition from a stable to unstable keyhole melting may be more nuanced than previously suggested[16,25] (discussed in detail later). We also noticed that the FKW remained relatively smooth at an approximately constant inclination angle, whereas the RKW presented random wrinkles and perturbations. With increasing AED, the keyhole penetration depth increases and the inclination of the FKW become steeper (higher FKW angle $\theta$, $\tan\theta \sim v_d/v_l$[39]), which is attributed to higher drilling velocity[40] ($v_d$) and increased energy coupling due to multiple reflections[8].

By supplementing our results with previous studies[16,25,40–42] across a wide range of powder materials, process conditions with different LPBF replicators and beamlines, we found that the FKW angle $\theta$ collapses to an inverse tangent of the normalised enthalpy product $(\Delta H/h_m \cdot L_{th}^*)$[37] (Fig.1b), where $L_{th}^*$ is the normalised thermal diffusion length[37] and the normalised enthalpy[6,43] $\Delta H/h_m$ a ratio of $\Delta H$, the deposited energy density[6] (also named as specific enthalpy) and $h_m$, the enthalpy at melting. This relationship is derived by the governing laws of heat transfer and kinematic equilibrium, elaborated in the "Methods" section. The agreement between the theorical derivation and experimental measurements (Fig.1b), suggests that the FKW inclination during LPBF is not only controlled by the deposited energy density $\Delta H$ and the material's melting enthalpy $h_m$, but also affected by the thermal diffusion length $L_{th}$.

Previous work has related the front keyhole wall to the laser fluence. Cunningham et al. [25] reported a nonlinear relationship

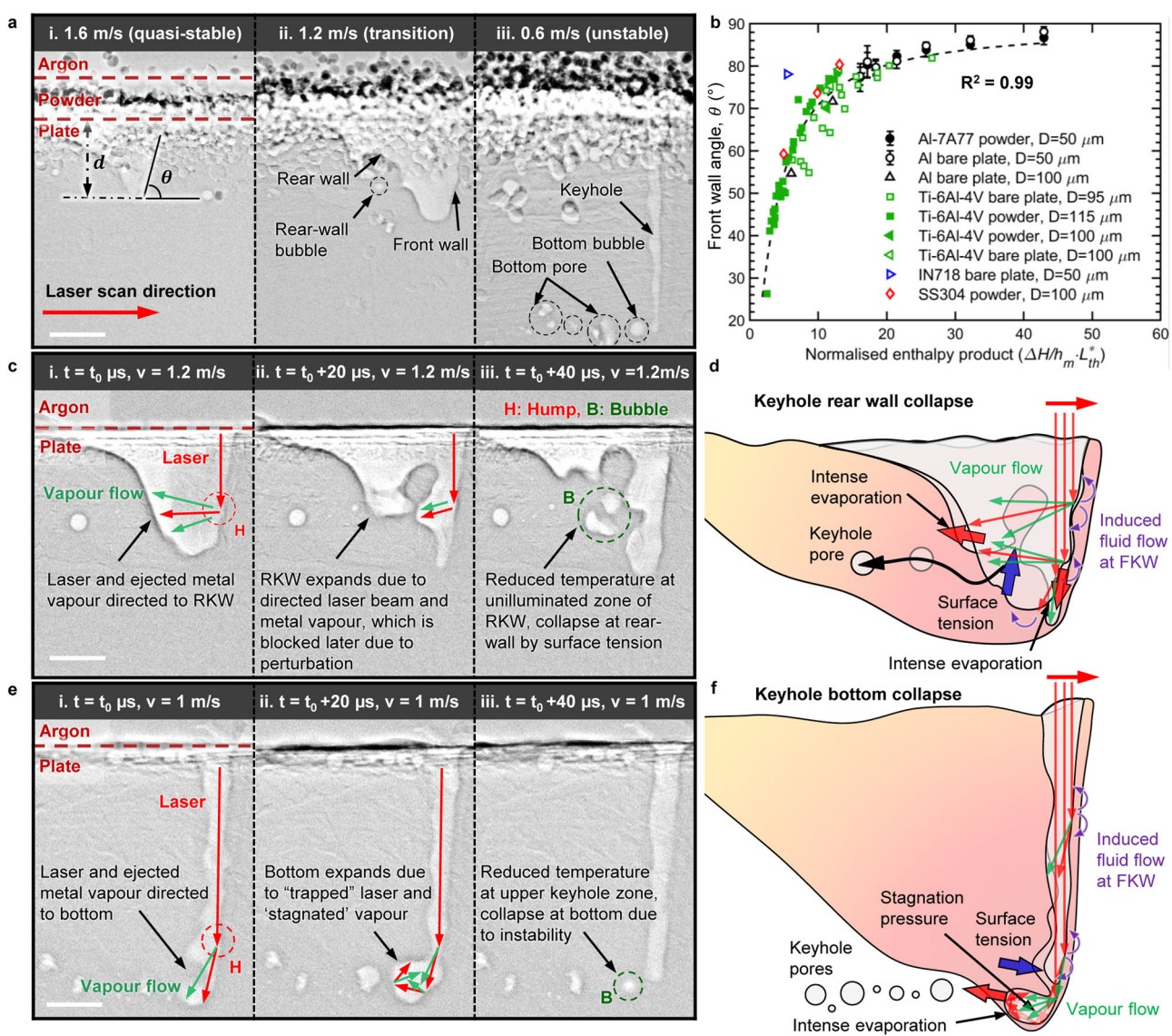

**Fig. 1 Keyhole collapse mechanism and related keyhole melting regime transitions in LPBF. a** Keyhole morphology variations from wide and shallow to narrow and deep across the (I) quasi-stable, (II) transition and (III) unstable keyhole regimes under different laser scan velocities. **b** Front keyhole wall (FKW) angle as a function of normalised enthalpy product for 9 datasets with four different materials. Curve fit is $\theta = \arctan\left[a \cdot \left(\Delta H/h_m \cdot L_{th}^* + b\right)\right]$ ($a = 0.29^{+0.04}_{-0.04}$, $b = -0.2^{+0.2}_{-0.6}$ with 95% confidence bounds), performed in Matlab using the Levenberg–Marquardt/least absolute residuals robust fitting algorithm. **c** Radiographs of laser melting with bare aluminium plate in (II) transition regime, showing rear keyhole wall (RKW) collapse with associated illustration (**d**). **e** Radiographs of laser melting with bare aluminium plate in (III) unstable regime, showing keyhole bottom collapse with associated illustration (**f**). $t_O$ is the time of the captured frame before the RKW or bottom keyhole expands. The red, blue and green arrows in **d** and **f** represent the laser beam, fluid flow and vapour flow, respectively. $d$ and $\theta$ represent the keyhole depth and FKW angle, respectively. Laser power 500 W, laser spot size 50 μm. All scale bars correspond to 150 μm. The datasets of LPBF with Ti-6Al-4V are cited from Cunningham et al.[25] (Fig. 4, S5 and S7) and Zhao et al.[16] (Movies S1–S5) with permission from AAAS. Datasets for LPBF with Inconel 718, SS 304 and aluminium bare plate are cited from Kouraytem et al.[40], Parab et al.[41], and Hojjatzadeh et al.[42], respectively.

between the FKW angle and the power density ($2P_l/\pi d_l^2$), which changes with the laser scan velocity as well as powder materials. Gan et al.[44] found that the tangent of FKW angle is approximately proportional to the "keyhole number Ke" (Ke $= \frac{1}{\sqrt{\pi}} \cdot \Delta H/h_m$), which is a scaled version of the normalised enthalpy. Here, we find even stronger agreement between the FKW angle and the normalised enthalpy product (Supplementary Fig. 3d), rather than the normalised enthalpy (Supplementary Fig. 3c). Our result builds on the work of Ye et al.[37], who first introduced the normalised enthalpy product in their scaling laws for keyhole depth (similar relations for keyhole depth measurements are shown in Supplementary Fig. 3a, b). The relationship

derived here also allows for defining thresholds between different melting regimes, similar to King et al.[6], who found the transition from conduction to keyhole melting occurs at a normalised enthalpy $\Delta H/h_m \approx (30 \pm 4)$ for 316L stainless steel.

Within the keyhole-melting regime, recent studies have reported a sharp transition between stable and unstable keyhole melting, typically defined by the onset of keyhole porosity[16,25,44]. From our data, we observed that the threshold for this transition can vary significantly between alloys. For Ti-6Al-4V[16,25] and Al7A77 (Figs.1b and 2d), we found this transition occurs at $\Delta H/h_m \cdot L_{th}^* \sim (8 \pm 3)$ or $\sim 60°$ FKW angle, and $\Delta H/h_m \cdot L_{th}^* \sim (20 \pm 3)$ or $\sim 80°$ FKW angle, respectively. The larger threshold

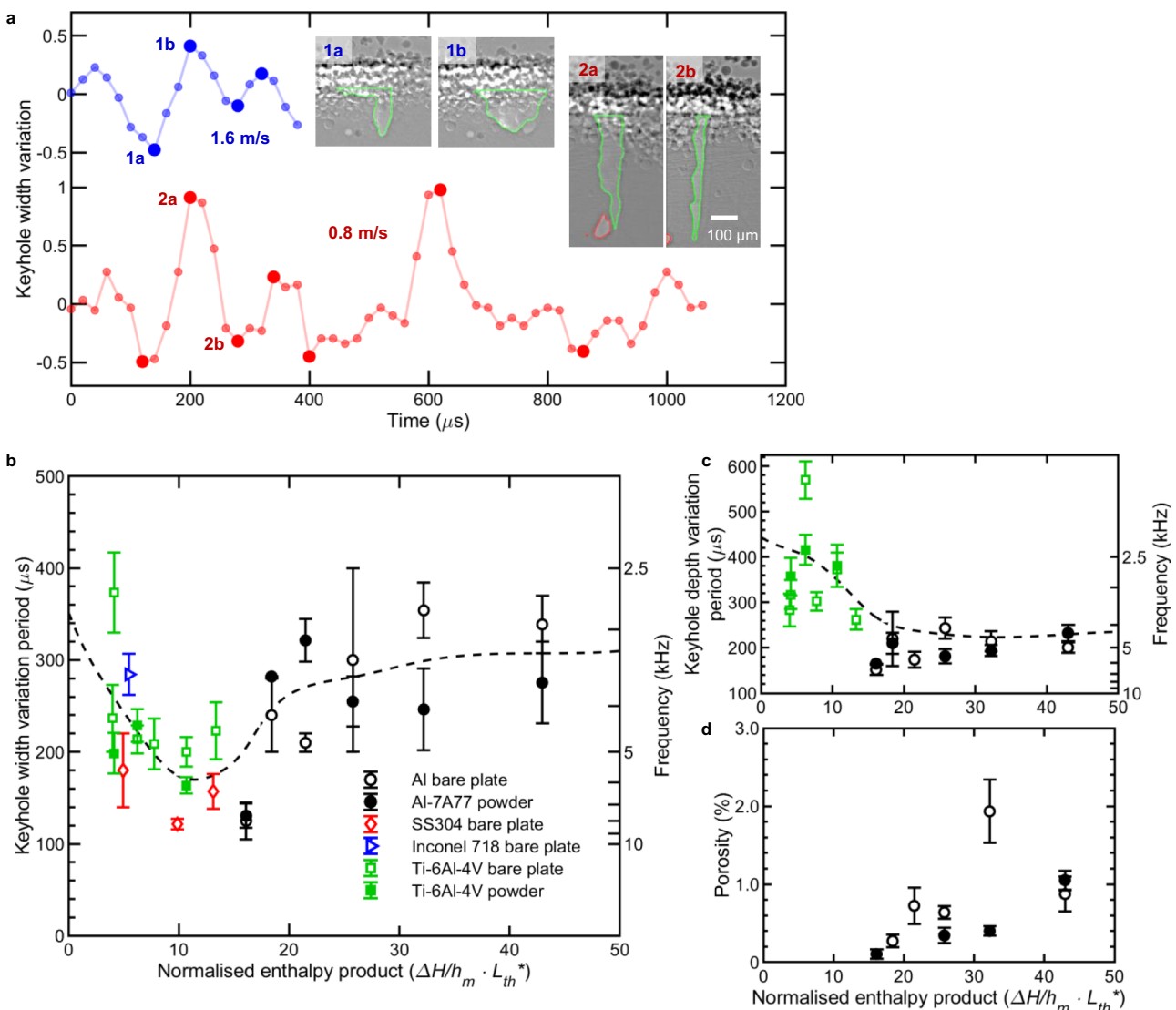

**Fig. 2 Keyhole dynamics in LPBF. a** Keyhole width relative to the mean, for laser scan velocities of 1.6 m/s (top, blue) and 0.8 m/s (bottom, red). Detected peaks/valleys are highlighted with marker size and example X-ray radiographs (1a, b; 2a, b). Average period between successive peaks/valleys in keyhole width **b** and depth **c** as a function of the normalised enthalpy product. Data for SS304, Inconel 718, and Ti64 are cited from references Kouraytem et al. [40], Parab et al. [41] and Zhao et al. [25, 16], respectively. Dashed lines are smoothing spline fits as a guide for the reader. **d** Percentage area porosity for cases with and without powder. Error bars represent standard deviation.

for Al7A77 is likely a combined result of its lower absorptivity at ambient temperature (~0.15 vs. ~0.45), larger Brewster angle (~85° [45] vs. ~80°, Supplementary Fig. 4), and lower melting enthalpy ($h_m = 2.63$ vs. $6.26$ J mm$^{-3}$).

In addition, we find that there can be an extended transition regime (II) between the stable (I) and unstable (III) keyhole regimes under high-power-velocity (high-$PV$). Pores begin to form in this transition regime, and initiate at the RKW rather than at the bottom of the keyhole (typical in III), which was also observed during laser welding of aluminium alloys[15] and low carbon steel[46], as well as LPBF of Ti–6Al–4V[42]. For similar AED, we found this transition regime becomes sharper with decreasing laser power and scan velocity ($P_l = 500$ W, $v_l = 1.4$ m/s, Supplementary Fig. 5a, c; $P_l = 200$ W, $v_l = 0.6$ m/s, Supplementary Fig. 5b, d), in agreement with Zhao et al. [16]. We speculate that the (II) transition regime is induced by the high-$PV$ combination under large AED, which enlarges the melt pool and vapour depression zone, leading to a relatively wider transition from the stable (I) to the unstable (III) keyhole regime. For the laser spot

size and alloy used in this study, a high-$PV$ with large AED is defined as $P_l = 500$ W, $v_l = 1.2$ m/s, AED ≥ 7 MJ · m$^{-2}$. The typical AED for LPBF is around 10 MJ m$^{-2}$ based on reference[14]. At high-$PV$ processing schemes for achieving large build rate in LPBF, we speculate that these schemes are most likely fall into this transition regime (II).

To further investigate the different keyhole collapse mechanisms in II and III, we compared the keyhole dynamics (Fig.1c, e and Supplementary Movies 1–12). "Humps" regularly form on the FKW due to the dependence of laser absorption on angle of incidence (Fresnel absorption[47], Supplementary Discussion 1), which becomes especially pronounced around the Brewster angle[47] (above which, absorptance falls off dramatically, Supplementary Fig. 4). In II (Fig.1c–i, AED = 8 MJ · m$^{-2}$, FKW angle 81.2 ± 1.7°), these humps tend to reflect the laser beam and the vapour flow towards the RKW. This leads to intensive evaporation and recoil pressure on the RKW and builds up a stagnation pressure[12], correspondingly deforming and expanding the RKW (Fig.1c–ii). Generally, the combined recoil and

stagnation pressure balances the surface tension acting on the free surface of the RKW, holding the overhanging RKW from collapse[14]. However, should the reflected laser beam and vapour flow be blocked or redirected by a perturbation of the keyhole (Fig.1c-ii), the surface temperature of the unilluminated RKW will quickly decrease. As the temperature decreases, surface tension increases linearly[5], overcoming the recoil pressure which decreases exponentially[5], causing a RKW collapse. We observed that this collapse can sometimes lead to the formation of bubbles from the RKW, approximately at the half-depth of keyhole (Fig.1c-iii), followed by the temporary formation of a deep, high aspect ratio depression. The melt flow at the middle of the pool half way up the RKW is still strong[48], as a result of the Marangoni-driven flow, propelling the bubble towards the rear of the melt pool, as discussed in detail later.

In the unstable regime (III) (Fig.1e-i, AED = 10 MJ m$^{-2}$ FKW angle 84.8 ± 0.8°), a narrow, deep keyhole forms, and humps on the FKW predominantly direct metal vapour and reflected laser beams to the bottom of the keyhole. Intense evaporation and recoil pressure at the keyhole bottom can be further amplified by the rapid formation of a vapour cavity ("J-shaped" keyhole, Fig.1e-ii), which traps reflected laser light and metal vapour, increasing the number and density of multiple reflections[8,20] and building up a significant stagnation pressure[12]. With energy concentrated in this cavity, the keyhole is prone to capillary instability and may sometimes collapse, pinching off a cavity to form a vapour filled bubble (Fig.1e-iii) and leads to a sharp decrease in keyhole depth. This is similar to, but not the same as the "spiking" as initially named in laser welding[11]. Spiking is also prevalent in LPBF but at turn-around points in raster scan patterns due to the finite acceleration of the laser beam, near-zero instantaneous scan velocity, and resulting pores at the "root" of keyhole[26]. While a small number of the bubbles we observed were re-captured by the expanding keyhole (Supplementary Fig. 2b), most were captured almost instantaneously by the advancing solidification front at the bottom of the keyhole to form pores in III.

**Keyhole radial and axial fluctuation and keyhole porosity**. To quantify the keyhole and bubble dynamics, we built an image processing pipeline (see the "Methods" section) to extract the keyhole depth and width from in situ X-ray radiographs (Fig.2a, Supplementary Fig. 6a, b). This was carried out for both our own study of LPBF with and without Al7A77 powder (Supplementary Movies 1–8 and 10–12), as well as a number of previous synchrotron X-ray studies[25,40,41] across different powder materials, process conditions, LPBF replicators, and beamlines (e.g., Parab et al. [41]). Note that the keyhole width is extracted as the median width along the whole keyhole depth.

Figure 2a shows the regular fluctuations in the keyhole width across different keyhole melting regimes (transition II, blue; unstable III, red). Similar, if not more regular, fluctuations were also observed without powder (Supplementary Fig. 6a, b). To further quantify these fluctuations, we calculated the average peak-to-peak period (see the "Methods" section) and found the corresponding frequencies of keyhole depth and width fluctuations range from ~2.5 to ~10 kHz, in agreement with previous acoustic, optical and radiometric measurements[8,49,50]. We also found significant trends in the keyhole width (Fig. 2b) and depth (Fig. 2c) fluctuations across different keyhole regimes: starting in I ($\Delta H/h_m \cdot L_{th}^* < 10$), the frequency of keyhole width fluctuations first increases, peaks in II ($10 < \Delta H/h_m \cdot L_{th}^* < 20$), and then slightly decreases in III ($\Delta H/h_m \cdot L_{th}^* > 20$). Similar patterns for the keyhole depth fluctuation are shown in Fig. 2c, which increases in frequency from I to II, and then remains high in III.

The keyhole width and depth fluctuation trends are consistent with the keyhole collapse mechanisms discussed above. In II, the high-frequency hump formation and subsequent migration down the FKW (Supplementary Discussion 1) can cause an open, wide vapour depression to temporarily collapse into a deeper, higher aspect ratio keyhole, with a significant decrease in keyhole width and increase in keyhole depth (although sometimes less significant), boosting the fluctuation frequency of keyhole. In III, relatively higher oscillation frequencies for the depth vs. the width also agrees with the discussion of Fig. 1, corresponding to bubbles being pinched off at the keyhole bottom, followed by a sharp decrease in keyhole depth. As shown in Fig. 2d, keyhole pores begin to form in II and increase in frequency through III. Comparing the final depth of pores relative to the substrate with the average keyhole depth (Supplementary Fig. 7) corroborates that bubbles initiate at the RKW in II and at the keyhole bottom in III.

Prior studies[16,25] reported larger keyhole fluctuations with powder compared to bare substrate. Zhao et al. [16] hypothesised that this phenomenon is induced by the momentary interaction between particle spatter and the laser beam[13], which shades the laser illumination and reduces recoil pressure, correspondingly increasing keyhole fluctuation. Here, by comparing the fluctuation frequency of keyhole width (Fig. 2b), depth (Fig. 2c), and also the tracked bubble numbers at per unit track length (Supplementary Fig. 8) with and without powder, we observed limited differences between the powder and bare plate samples. We hypothesise that the shadowing effect of particle spatter on the laser beam is less significant when a high laser power and a thin powder layer thickness are applied (for the laser spot size and alloy used in this study, a high laser power and a thin layer thickness is defined as ≥500 W and ≤30 μm, respectively), which is consistent with the finding reported by Khairallah et al. [13]. Khairallah et al. found that there exists a power threshold beyond which the particle spatter expulsion mechanism is activated and could vaporise the spatter quickly, inversely, inducing pores due to laser shadowing with rapid cooling.

**Keyhole-induced bubble lifetime dynamics in LPBF**. Using our image processing pipeline (e.g., Kalman filter tracking[51]), we traced the evolutions of the keyhole-induced bubbles and extracted their centroids and equivalent diameters over their lifetime, starting after a bubble is pinched off from the keyhole and ending when the bubble is fully captured by the solidification front (see examples in Fig. 3, Supplementary Fig. 9a and Supplementary Movies 7, 8 with AED = 10 MJ m$^{-2}$; Supplementary Figs. 9b, 10 and Supplementary Movies 1–4 with AED = 17 MJ m$^{-2}$). We observed that bubbles evolve through three main stages, with and without the presence of a powder layer:

(1) bubbles rapidly grow immediately after being pinched off from the keyhole (Fig. 3a–ii, iii, b–ii, iii, Fig. 3c, Supplementary Fig. 10d), thought to be due to pressure equalisation; then

(2) the bubbles shrink while migrating towards the rear side of the melt pool (Fig. 3a–iv, v, b–iv, v Supplementary Fig. 10a–ii, iii, b–ii, iii, c), hypothesised to be caused by the condensation of the metal vapour in them, competing with the diffusion of hydrogen into the bubbles; and finally

(3) they are captured by the advancing solidification front (Fig.3a–vi, b–vi, Supplementary Fig. 10a–v, b–v).

In stage (1), as the bubble was pinched off from the keyhole, the bubble inner pressure $p_i$ is expected to be similar to the keyhole bottom recoil pressure ($\sim 10^5 - 10^6$ Pa[14,40]), which is generally larger than the ambient pressure $p_a$ ($\sim 1 \times 10^5$ Pa). This

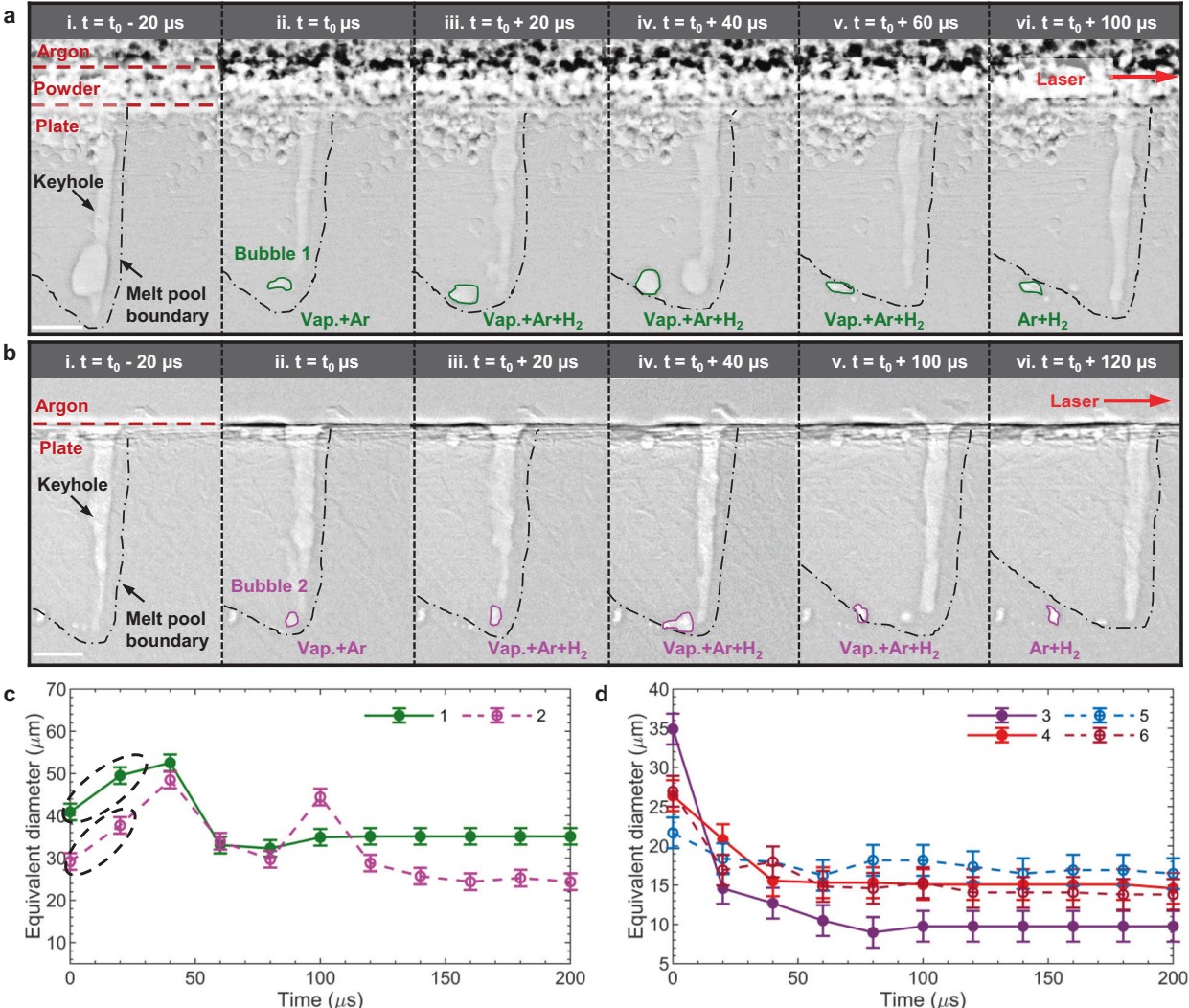

**Fig. 3 Keyhole bubble lifetime dynamics during LPBF.** Laser scan velocity 1 m/s and laser power 500 W. **a** and **b** are radiographs with Al7A77 powder and bare aluminium plate, respectively. **c** and **d** show example time evolutions of bubble equivalent diameter during LPBF with (solid line) and without (dash line) Al7A77 powder, respectively. The equivalent diameter is calculated using $\sqrt{6A/\pi}$, where $A$ is the bubble area measured from the X-ray image. Note the bubble size error is calculated as ±2 pixels (1.96 μm/pixel), equivalent to the segmentation uncertainty. The total tracked bubble numbers are 5 and 8 for the powder and bare plate cases (Supplementary Fig. 9a), respectively, using a criterion where the minimum number of frames that a bubble is identified is 6 (see the "Methods" section). The time $t_0$ is set to the moment a bubble is first identified (Note, $t_0$ is set as $t_0 = 0$ in **c** and **d**). The black dashed circles show initial bubble growth in **c**. The bubbles of interest shown in **a** and **b** are marked by green and lavender colours, respectively, corresponding to same colours in **c**. Vap. vapour, Ar argon, $H_2$ hydrogen. All scale bars correspond to 100 μm.

pressure difference then drives bubble growth according to the ideal gas law[52] ($p = nRT/V$), where the volume, $V$, must increase to accommodate the reduction in pressure $p$ from $p_i$ to $p_a$. (Note, $n$ is the molar number of gas, $R$ the universal gas constant, and $T$ the temperature). Simultaneously, as the surrounding liquid metal cools the bubble, the superheated metal-vapour inside the bubble will condense, reducing $n$, and hence decreasing the bubble volume $V$, but at a slower rate than the pressure equalisation (discussed in detail later). This is also known as the bubble contraction mechanism in laser welding[15,53].

In stage (2) bubbles shrink while migrating towards the rear side of melt pool, we observed that the bubble shrinkage undergoes a marked slowdown at the later stage of condensation (e.g., bubble 3 from 40 to 120 μs in Fig. 3d, bubble 1 from 40 to 80 μs in Supplementary Fig. 10c), and the bubble size then get stabilised. We speculate the reduction in shrinkage rate and eventual bubble size stabilisation are caused by the hydrogen

diffusion[34]. The presence of hydrogen in keyhole pores was observed by Matsunawa et al. [15], who measured ~3–12% hydrogen content in pores formed during laser welding of aluminium alloy using mass spectrometry. Hydrogen is expected to be present in both the virgin substrate and powder particles. During LPBF, the melt at the advancing solidification front can then become supersaturated with hydrogen, driving hydrogen diffusion from the melt into the bubble[30,32] and it is several orders faster than the diffusion of other atoms[54].

In stage (3) as bubbles interact with the solidification front, we observed that they experience sudden bursts of growth and shrinkage (e.g., bubble 2 at 80–120 μs in Fig. 3c, Supplementary Fig. 11a, b). This phenomenon may be explained by the interaction of the bubble with the rapidly growing solidification microstructure[55], where the cells and dendrites can restrict and distort the bubbles (Supplementary Fig. 11c), forming complex non-spherical pores, as described in Supplementary Discussion 2.

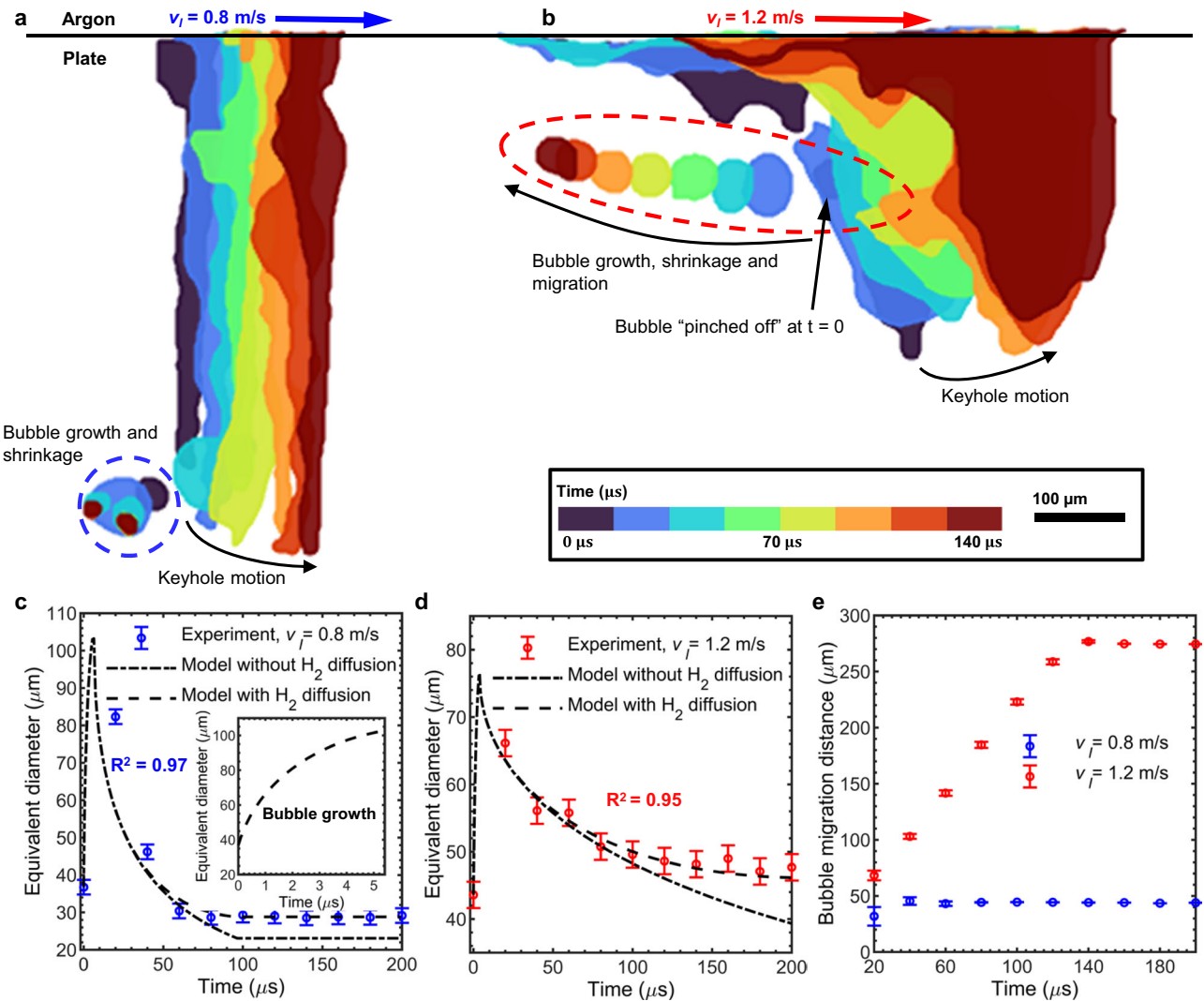

**Fig. 4 Tracking and modelling of keyhole induced bubble dynamics.** Colour map tracking for keyhole and bubble under low **a** and high **b** laser scan velocities, corresponding to regimes (III) and (II), respectively. Comparing the modelled bubble size variations with in situ X-ray measurements at low **c** and high **d** laser scan velocities. The equivalent diameter is calculated using $\sqrt{6A/\pi}$, where $A$ is the bubble area measured from X-ray image (see the "Methods" section). The bubble size error is calculated as ±2 pixels (1.96 μm/pixel), equivalent to the segmentation uncertainty. Note, the bubble shown in **a** split into two small ones in the later stage, where the equivalent diameter is estimated based on their sum area in (**c**). The temporal resolution of X-ray imaging (20 μs) is insufficient to capture the whole process of bubble growth, therefore, we are unable to get enough data and fully verify the bubble growth model. **e** Bubble migration distance compared to their initial formed location. The bubble migration distance error is calculated based on the bubble motion with instantaneous speeds (0–5 m/s) during the finite camera exposure time (2.5 μs). Low laser scan velocity 0.8 m/s, high scan velocity 1.2 m/s, laser power 500 W. The time 0 is set when a bubble first appears.

Occasionally, other bubble dynamics were observed, including re-captured by the expanding keyhole, coalescence, and even splitting (Supplementary Movies 1–12).

Based on the above findings, from the initiation of a bubble until it gets frozen as a pore, its composition will initially be a combination of metal vapour and shielding gas argon (Ar), which is driven into the keyhole via the Bernoulli effect[56]. The metal vapour will condense, leaving the Ar, and reducing the bubble size. Simultaneously some hydrogen ($H_2$) will diffuse in, slowing the bubble shrinkage. These stages are highlighted by the tracked bubble colours in Fig. 3a, b. Note that the argon can be treated as insoluble in molten aluminium[57], and is therefore expected to be the major content left in the frozen pore.

To verify our hypothesis in the above discussion, we combined the Rayleigh–Plesset equation[58], bubble condensation model from Florschuetz and Chao[59], and the ideal gas law[52] to build a united bubble model (see the "Methods" section) while considering

pressure-driven growth, vapour condensation and hydrogen diffusion. We compared the modelled results with experimental measurements under different keyhole melting regimes (III, II), shown in Fig. 4, where we presented the tracked keyhole and keyhole pore transient trajectories at 0.8 and 1.2 m/s laser velocities (Fig. 4a, b) based on the X-ray images (Supplementary Movies 5, 6, 10, 11), corresponding to III (AED = 12.5 MJ m$^{-2}$) and II (AED = 8 MJ m$^{-2}$).

As seen from Fig. 4c, d, immediately after forming the bubble experiences explosive growth, which plateaus after about ~3–5 μs (Supplementary Table 1). The bubble then begins to shrink at a slower rate, and eventually stabilises (~50–150 μs), being consistent with the experimental measurements. (Note that deviations may be caused by the effect of surface tension and fluid flow that are not included in the built model.) We hypothesise that the explosive growth is caused by the pressure equalisation, with the bubble volume increasing like $t^3$ (see the "Methods" section). The

decline and plateau in bubble growth is explained by the decrease in driving force $(p_i - p_a)$ with increasing bubble volume, based on the ideal gas law. We also speculate that the bubble shrinkage rate is slower than the bubble growth rate, proportional to $t^{-1/2}$ and $t$, respectively (see the "Methods" section). Additionally, we hypothesise that the bubble shrinkage rate can be further slowed by the diffusion of hydrogen. The modelling results (Fig. 4c, d) show that accounting for the hydrogen diffusion is necessary to explain the latter stage of bubble size stabilisation and this leads to good agreement with the tracked bubble equivalent diameter. Note that our temporal resolution of X-ray imaging (20 μs) is insufficient to capture the whole process of bubble growth after being pinched off from the keyhole. With a calculated growth time on the order of microseconds, it is likely that we often fail to capture this growth, explaining why immediate bubble shrinkage is observed more frequently than initial bubble growth then shrinkage (Supplementary Fig. 9a, b).

The bubbles migrated a larger distance in II with a tendency of going backward and upward, when initiated at the RKW, than in III (Fig. 4e), when initiated at the bottom of the keyhole, remaining almost stationary. This is due to two effects (i) the proximity of the solidification front to the bottom of the keyhole, bubbles having insufficient time to move upwards in the melt pool[20]; and (ii) the melt flow velocity and flow pattern are location dependent across the melt pool[48], and the induced drag force is local flow velocity dependent. The flow velocity is higher in the middle of the pool, hence when a bubble is formed half way along the RKW, it quickly flows backwards to the rear of melt pool. At the bottom of the pool near the solid–liquid interface, the flow velocity is low, hence bubbles detaching from the very bottom of keyhole remain nearly stationary until captured by the solidification front and turn to pores. The bubble in Fig. 4a (regime III) has an average velocity of $1.0 \pm 0.5$ m/s (measured over the first 60 μs), while the bubble in Fig. 4b (regime II) moves at $2.4 \pm 0.7$ m/s. In regime III with high AED, the pore migration speed of 1.0 m/s agrees well with our previous measurements of the Marangoni flow speed with tungsten carbide particles (0.97 m/s under the same AED)[60]. In regime II at lower AED, we assume that a strong viscous drag force is responsible for the higher initial speed of bubbles initiated at the RKW.

## Discussion

In summary, this manuscript reveals the lifetime dynamics of keyhole pore (growth, shrinkage, migration, interaction with solidification microstructure and capture by advancing solidification front), introducing a threshold, the normalised enthalpy product, to reveal and elucidate different keyhole pore generation mechanisms and their corresponding keyhole melting regimes under stable, transition and unstable conditions in LPBF. Our findings on keyhole fluctuation and bubble dynamics provide critical guidance (e.g., bubble growth/shrinkage rate, pore location and size) to achieve in situ pore elimination by remelting[27,61] using dual-laser LPBF machines[62] or hybrid LPBF[63], and pore suppression via real-time control of keyhole dynamics (e.g., beam oscillation[64]) in a broad range of high-energy-beam processing techniques (e.g., electron beam melting[65], keyhole laser welding[64] and laser drilling[66]).

In this study, we combined our in situ synchrotron X-ray imaging results from LPBF of an aluminium alloy Al7A77 with recent studies of other key additive manufacturing alloys (e.g., Ti–6Al–4V, Inconel 718, SS 304). We found a transition regime (II) between the stable (I) and unstable (III) keyhole regimes. This transition regime (II) is most pronounced for high-$PV$ combinations with large AED (AED $\geq 7$ MJ · m$^{-2}$). As shown in Fig. 1, the vapour depression becomes unstable in regime (II), randomly

**Table 1 Process parameters for the LPBF experiments.**

| Parameters | Values | Parameters | Values |
|---|---|---|---|
| Laser scan velocity [m s$^{-1}$] | 0.6, 0.8, 1, 1.2, 1.4, 1.6 | Laser power [W] | 200, 500 |
| Track length [mm] | 5 | Laser spot size [μm] | 50 |
| Layer thickness [μm] | 30 | Aluminium substrate size [mm] | 46 × 17 × 0.5 |

collapsing and inducing pores at the middle of the rear keyhole wall (RKW), as opposed to at the bottom of the keyhole in regime (III), which is the traditionally observed location for pore formation. We also observed significant trends in keyhole fluctuation frequency (radial and axial) across the different keyhole regimes, with the fastest fluctuation occurring in the transition regime (II) at ~10 kHz (Fig. 2). Based on our observations, we developed a material, machine and process condition agnostic relationship for the front keyhole wall (FKW) angle, which collapses to a single function of the normalised enthalpy product (Fig. 1b). The resulting relation provides a non-dimensional threshold for predicting the three keyhole regime transitions and the onset of keyhole porosity for different alloys and processing conditions (e.g., laser spot size, laser power, laser scan velocity).

In addition, we clarified the keyhole pore formation process, including the lifetime dynamics of vapour bubbles in the melt pool, which is characterised by three stages (Fig. 3): (1) fast pressure-driven growth, (2) shrinkage by metal vapour condensation, slowed by hydrogen diffusion, and (3) interaction with solidification microstructure (e.g., cellular-dendrites) and capture by the advancing solidification front. Furthermore, we proposed a model of bubble growth and shrinkage (Fig. 4), including the physics of pressure-driven growth, vapour condensation and hydrogen diffusion. This model was found to be consistent with the experimental data, supporting our hypotheses: (i) explosive bubble growth during the early lifetime of stage (1) is mainly a pressure-driven process, where the bubble volume expands like ~$t^3$; and (ii) hydrogen diffusion is sufficiently high to stabilise the bubble size at the later stage of condensation in stage (2).

## Methods

**LPBF replicator (ISOPR) system and processing conditions**. In situ synchrotron X-ray imaging was performed at the Argonne National Laboratory's Advanced Photon Source (APS) to probe keyhole and bubble dynamics during LPBF using ISOPR (Supplementary Fig. 1). ISOPR was custom-designed to accommodate synchrotron X-ray imaging of the LPBF process and includes a continuous wave Ytterbium-doped fibre laser (IPG YLR-500-AC, USA) with a wavelength of $1070 \pm 10$ nm and maximum power of 520 W, an $X$–$Y$ galvanometer scanner (intelliSCANde 30, SCANLAB GmbH, Germany), an environmental chamber and a sample holder positioned at the centre of the chamber. During the experiment, the chamber was filled with argon gas at a pressure of +10 kPa to reduce oxidation. The keyhole and pores were imaged at high spatial (1.96 μm) and temporal (frame rate 50 kHz) resolutions with a FASTCAM SA-Z 2100K (Photron, USA) camera by converting the attenuated X-ray beam to optically visible light using a 100 μm-thick LuAg:Ce scintillator. The resultant field of view was 512 pixels (1 mm) in width by 680 pixels (1.33 mm) in height. The commercially Al7A77 powder (HRL Laboratory, USA, material composition shown in Supplementary Table 2) with a particle size range of 15–45 μm, and pure aluminium (Goodfellow, UK) plate with purity of 99.99% sandwiched between two 1 mm thickness glassy carbon plates (HTW, Germany), were used in this study with the process parameters shown in Table 1. Thermophysical properties of the materials are listed in Supplementary Table 3.

**Dimensionless analysis on front keyhole wall angle**. The FKW angle is determined by its equilibrium kinematic condition, described as $\tan\theta = v_d/v_l$, which can also be estimated based on keyhole geometry as in reference[67], $\tan\theta \approx d/d_l$ ($d$ is the keyhole depth). Linear relations between $\tan\theta$ and the keyhole depth $d$ have been verified by recent studies[16,25] over different materials in LPBF. Note, Zhao et al.[16] found that the ratio $d/\tan\theta$ can fluctuate around an effective laser spot size and falls with decreasing scan velocity. Here, we assume that the effective laser spot

size might be approximated by the original laser beam diameter and the FKW angle may be expressed as, $\tan\theta \approx d/d_l$, the keyhole depth $d$ is approximately equal to the melt pool depth $d_m$ with any difference being incorporated into the coefficient $K_1$ as[44], $d \approx K_1 \cdot d_m$. The melt pool depth is estimated as the distance that the melt front advances in the axial direction during the dwell time ($r_l/v_l$, where $r_l$ is the laser beam radius), in the form of Eq. (1) based on the governing laws of heat transfer[37],

$$d_m = K_2 \cdot \left(\frac{\triangle H}{h_m}\right) L_{th} \tag{1}$$

the FKW angle can therefore be expressed as a function of the normalised enthalpy product,

$$\theta \approx atan\left(K_3 \cdot \frac{\triangle H}{h_m} \cdot L_{th}^*\right) \tag{2}$$

where $K_2$ is a coefficient to account for differences between the actual and modelled melt pool dimensions, $K_3 = K_1 \cdot K_2/2$, $L_{th} = \sqrt{(\alpha_l \cdot r_l)/v_l}$ the thermal diffusion length, $\Delta H/h_m = (\beta \cdot P_l)/\left(h_m\sqrt{\pi \cdot \alpha_l \cdot v_l \cdot r_l^3}\right)$ the normalised enthalpy[6,43] (Note, $\beta$ is the laser absorptivity, $\alpha_l$ is the liquid thermal diffusivity, $h_m = \rho_l \cdot c_l \cdot T_l$ the enthalpy at melting with $\rho_l$ the density, $c_l$ the heat capacity at melting temperature $T_l$) and $L_{th}^* = L_{th}/r_l$ is the normalised thermal diffusion length.

**Image and data processing pipeline.** The major steps of our image processing pipeline (Supplementary Fig. 12) are as follows:

i. Flat field correction, subtracting the offset background, followed by a 2D Gaussian filtering (Supplementary Fig. 12a, b). Note that the flat field correction used a general equation[28]: FFC = ($I_0$−Flat$_{avg}$)/(Flat$_{avg}$−Dark$_{avg}$), where FFC, $I_0$, Flat$_{avg}$ and Dark$_{avg}$ represent the flat field corrected image, raw image, average of 100 flat filed images and average of 100 dark field images, respectively.

ii. Initial image segmentation (Supplementary Fig. 12c) with $K$-means clustering algorithm[68]. The segmentation uncertainty is around ±2 pixels (1.96 µm/pixel).

iii. Frame stack time-domain integration (Supplementary Fig. 12d) by applying volume threshold for noise reduction.

iv. The final segmented keyhole and keyhole bubble were achieved by applying (iii) over the whole radiograph stack. Examples are shown in Supplementary Fig. 13a-iii, b-ii and Supplementary Movies, 2, 4, 6, 8, 11, where the keyhole and keyhole bubble were marked with green and red colours, respectively.

v. Kalman filter tracking algorithm[51] was developed based on the segmented keyhole bubble in step (iv). The minimum number of frames that a bubble is identified was set as 6 for being considered as an effective track. Bubbles observed for less than 6 frames were mainly recaptured by the keyhole.

Note that the segmentation algorithm developed in step (ii) fails in very limited cases due to background fluctuations (e.g., particles randomly falling into the gap between the glassy carbon and aluminium plates). Here we also developed a supervised machine learning (random decision forests) model to segment keyhole and bubbles (Supplementary Fig. 13b-iii), but found that the machine learning model fails more often during segmentation, possibly induced by the non-perfect ground-truth labelling as the X-ray frames may have different background fluctuations. The Kalman tracking algorithm developed in step (v) also fails occasionally due to the splitting or coalescence of bubbles, as well as background fluctuations. However, these segmentation and tracking errors are minor and do not significantly affect the quantified keyhole and bubble dynamics.

**Peak-to-peak period for keyhole width/depth.** Matlab's built-in findpeaks function[69] was used to extract peaks and valleys from the keyhole width/depth signatures. To mitigate the effect of outliers and noise, the signatures were pre-processed using a moving mean filter with a width of 3. A linear fit of the data was also subtracted to remove slow changes in the width/depth. To remain safely below the Nyquist frequency (25 kHz), the minimum peak distance was set to 5 (100 µs), acting as a simple low pass filter with cut-off frequency of 10 kHz. The minimum peak prominence was set to 1 standard deviation of the keyhole width/depth data set.

**Bubble growth model with condensation and hydrogen diffusion.** In the LPBF process, a rigorous modelling of the bubble dynamics while coupling the mesoscale and nanosecond multi-physics of the process remains a challenge. Here, we assume that the contents, temperature and pressure within the bubble are homogeneous, the bubble remains spherical and at rest relative to the incompressible melt pool flow. We take the bubble equivalent radius that the bubble is first identified as the bubble initial radius $r_{b0}$ at the moment $t = 0$. In stage (1), considering the very short time of bubble growth, we suppose that there is negligible mass or thermal transfer over the bubble interface (no condensation or gas diffusion). Accordingly, the instantaneous bubble radius $r_b(t)$ at time $t$ can be described by the

Rayleigh–Plesset equation[58],

$$r_b \cdot \frac{\partial}{\partial t}\left(\frac{\partial r_b}{\partial t}\right) + \frac{3}{2}\left(\frac{\partial r_b}{\partial t}\right)^2 = \frac{1}{\rho_l}\left(p_i - p_a - \frac{2\sigma}{r_b} - 4\mu_l \cdot \frac{\partial r_b}{\partial t}\right) \tag{3}$$

where $\sigma$ is the surface tension and $\mu_l$ is the viscosity.

Since the surface tension and the viscosity terms ($\sim 10^4$ Pa) are both negligible compared to the pressure difference $p_i - p_a$ ($\sim 10^5 - 10^6$ Pa)[14,40], which can be omitted in the above Rayleigh–Plesset equation, the bubble growth rate can be approximated as[58]

$$\frac{\partial r_b}{\partial t} \to \left(\frac{2}{3} \cdot \frac{p_i - p_a}{\rho_l}\right)^{\frac{1}{2}}, r_b(t) \gg r_b(0) \tag{4}$$

This derived bubble growth rate suggests that, early in stage (1) with maximum pressure difference, the bubble grows with a function of time $t$, while the bubble volume expands like $t^3$. The initial bubble inner pressure $p_i(0)$ may be approximated as the recoil pressure $p_{recoil}$, $p_i(0) \approx p_{recoil}$, where the recoil pressure is a function of temperature $T$ based on Anisimov's evaporation model[5,14],

$$p_{recoil} = 0.54p_a\exp\left[\frac{\lambda}{K_B}\left(\frac{1}{T} - \frac{1}{T_v}\right)\right] \tag{5}$$

where $\lambda = 293.4$ kJ mol$^{-1}$ is the latent heat of evaporation per atom of aluminium, $K_B = 8.314\times 10^{-3}$ kJ mol$^{-1}$ $K^{-1}$ the Boltzmann constant, $T$ the keyhole surface temperature, $T_v = 2753.15$ K is the evaporation temperature of aluminium. By using the 2D moving heat source model[6], the keyhole surface temperature is approximated by the melt pool peak temperature,

$$T = \frac{\sqrt{2}\beta Ir_l}{k_l\sqrt{\pi}}\tan^{-1}\sqrt{\frac{2a_l}{v_lr_l}} \tag{6}$$

where $I$ is the laser intensity (approximated as $I = P_l/\left(\pi r_l^2\right)$), and $k_l = \rho_l \cdot c_l \cdot a_l$ is the liquidus thermal conductivity.

By combining the above Eqs. (4)–(6) and the ideal gas law[52] ($p = nRT/V$), the transient bubble size in stage (1) can be calculated by $r_b(t) = r_{b0} + \triangle r_{b1}(t), t \leq t_1$, where $\triangle r_{b1}(t)$ (solved from simultaneous Eqs. (4)–(6) and the ideal gas law) is the pressure-driven increment in bubble radius at time $t$, and $t_1$ is the pressure-driven bubble growth time. Note that $t_1$ is calculated as the time that the pressure difference $p_i - p_a$ reduces to a percentage threshold of $p_a$ (we used 5% and defined as $p_i(t_1) - p_a \leq 0.05p_a$).

In the condensation dominated stage (2), Florschuetz and Chao[59] built the following relation between the bubble instantaneous radius $r_{b2}(t)$ and the radius at the beginning of vapour condensation $r_{b2}(0)$ (approximated as the $r_b(t_1)$),

$$\frac{1}{3}\left[\frac{r_{b2}(t)}{r_{b2}(0)}\right]^2 + \frac{2}{3}\frac{r_{b2}(0)}{r_{b2}(t)} = 1 + \frac{t}{t_{cond}}, t_{cond} = \frac{\pi\left[r_{b2}(0)\right]^2}{4Ja^2a_l} \tag{7}$$

where $t_{cond}$ is the condensation characteristic time and Ja is the Jakob number, Ja $= \frac{\rho_lc_l(T_{bs}-T_{sat})}{\rho_vL_v}$ ($T_{bs}$ is the bubble surface temperature, $T_{sat}$ is the saturated temperature, $L_v = 1.02\times 10^7$ J kg$^{-1}$ is the latent heat of evaporation and $\rho_v = 1850$ kg m$^{-3}$ is the vapour density). In stage (2), the dissolved hydrogen in melt pool may diffuse into the bubble driven by the concentration difference. The bubble size $r_{b3}$ induced by hydrogen diffusion may be estimated by the characteristic length of hydrogen diffusion limited growth $l_D$[35] as, $r_{b3} = l_D/2, l_D = \sqrt{D_ht}$, where $D_h$ is the mass diffusivity of hydrogen in liquidus aluminium and may be approximated as an average of $D_h(T_l) = 1.0943\times 10^{-7}$m$^2$ s$^{-1}$ and $D_h(T_v) = 1.1302\times 10^{-5}$m$^2$ s$^{-1}$ based on ref.[70]. Accordingly, the bubble radius in stage (2) that is controlled by vapour condensation and hydrogen diffusion may be described as $\frac{4}{3}\pi r_b^3 = \frac{4}{3}\pi r_{b2}^3 + \frac{4}{3}\pi r_{b3}^3$, based on mass balance (assuming that the contents, temperature and pressure within the bubble are homogeneous). Therefore, the instantaneous bubble radius in stages (1) and (2) can be approximated as

$$r_b(t) = \begin{cases} r_{b0} + \triangle r_{b1}(t), t \leq t_1 \\ \sqrt[3]{\left[r_{b2}(t-t_1)\right]^3 + \left[r_{b3}(t-t_1)\right]^3}, t > t_1 \end{cases} \tag{8}$$

All the parameters used in the model were listed in Supplementary Table 1 (characteristic parameters) and Supplementary Table 3 (thermal–physical properties), which were used to plot the bubble diameter graph in Fig. 4c, d. Note that we used the liquidus temperature for aluminium $T_l = 933.5$ K to approximate the bubble saturation temperature $T_{sat}$ in the melt pool, while for the bubble surface temperature $T_{bs}$, we approximated its magnitude by fitting between observed data and the modelled results (Note, the bubble is subcooled in the melt pool with uncertainty temperature change under unknown time period after it has being pinched off from keyhole). For a spherical vapour bubble, the total condensation time is around $1 \sim 3t_{cond}$[32]. Here, we used the average $2t_{cond}$ and modelled the bubble dynamics within $2t_{cond}$.

## Data availability

Representative data that support the findings are given in the figures and tables (main paper and Supplementary Information). Other datasets are available from the corresponding author on reasonable request.

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

## Acknowledgements

This research is supported by the Office of Naval Research (ONR) Grant N62909-19-1-2109 to P.D.L., Y.H. and S.J.C., the Engineering and Physical Sciences Research Council (EPSRC) grants (EP/R511638/1 and EP/V061798/1) for P.D.L. and C.L.A.L., EPSRC grant (EP/N509577/1) for C.L.A.L., EPSRC grant (EP/T001569/1) via the Alan Turing Institute for J.T., the UK-EPSRC via MAPP: EPSRC Future Manufacturing Hub in Manufacture using Advanced Powder Processes (EP/P006566/1) for P.D.L., C.L.A.L., Y.H., T.G.F., S.J.C. and S.M., and the Royal Academy of Engineering (CiET1819/10) for P.D.L. This research used resources of the Advanced Photon Source, a U.S. Department of Energy (DOE) Office of Science User Facility, operated for the DOE Office of Science by Argonne National Laboratory under Contract No. DE-AC02-06CH11357. We also acknowledge the use of facilities and support provided by the Research Complex at Harwell, and the Advanced Photon Source for providing the beam-time (213874). We are grateful for HRL laboratory for providing the Al7A77 powder for this study. The authors also acknowledge the beamline experiment support from Yunhui Chen, Lorna Sinclair, David Rees, Niranjan Parab and other staff at the 32-ID beamline for their assistance, and sample preparation support from Elena Ruckh and Saurabh Shah for tomographic scans.

## Author contributions

P.D.L. and H.Y. conceived the project. H.Y. wrote the manuscript with significant contributions from P.D.L., C.L.A.L., T.G.F., and S.J.C., performed the characterisation, image processing (with help from C.L.A.L. and J.T.), performed data analysis and modelling. P.D.L., C.L.A.L., S.J.C., S.M. and K.F. led and performed the experiments. T.G.F. calculated the peak-to-peak period and frequency for keyhole width/depth. S.J.C. programmed the Kalman filter tracking.

## Competing interests

The authors declare no competing interests.
