## [Peer Review File · Nature Communications]

Title: Keyhole fluctuation and pore formation mechanisms during laser powder bed fusion additive manufacturingREVIEWER COMMENTS

Reviewer #1 (Remarks to the Author):

Keyhole fluctuation and porosity is an important question in additive manufacturing. This study provides more detailed and deeper insights into the bubble evolution before forming keyhole pores, and some conclusions are generally applicable to different materials, which is a significant step forward. While the manuscript is worthy of publication, the following comments are for the authors' consideration to further improve the manuscript:

1) In abstract, specifying the radial keyhole fluctuations to be ~ 10 kHz is misleading, because the frequency is $2.5\sim 10$ kHz in the results.

2) In Supplementary Line 93, the variable u should be v_l (laser scan speed).

3) In Supplementary Line 104, v_l may not be proper to be used in the Péclet number equation, and instead it should be the local flow velocity, which is usually higher than the laser scan speed but approximately at the same order though, because the physical meanings are different. Thus, it is not necessary to relate the mode of thermal transfer with the keyhole morphology in Line 124 of the manuscript.

4) It is interesting to see the good agreement between the theoretical inverse tangent relationship and the experimental results. There are several questions about this. i) Can the normalized enthalpy product distinguish whether there is a keyhole for different materials? ii) As we know that the ambient pressure influences the keyhole, is it possible to further consider this factor in the derivation? iii) I might miss it: what are the values of a , b and c ? While the physical meaning of a seems obvious, what are the physical meanings of b and c ?

5) In Line 133, "the larger threshold for Al7A77" should be "smaller"? Because the value is ~ 8 for Al7A77 vs ~ 20 for TC4.

6) For the keyhole fluctuation period and frequency in Fig.2, would it be better to perform Fourier analysis to extract the significant patterns? As there are multiple forces acting on the keyhole, they may have different patterns and thus induce different fluctuation frequencies.

7) In Fig.2b, it may be improper to claim the shaded blue as the transition regime for all the four alloys there, because the authors claimed the threshold for the stable/unstable keyhole transition can vary significantly between alloys.

8) It is good to see the good agreement between the model and experiments on the bubble size evolution. However, hydrogen diffusion still has no direct experimental evidence, and the model has not been validated in similar scenarios. Better to achieve some experimental evidence (maybe atom probe tomograph can detect the hydrogen on the keyhole pore surfaces) or use some models that have been

validated in additive manufacturing scenarios.

Reviewer #2 (Remarks to the Author):

This article provides a detailed study of how porosity develops in the laser powder-bed fusion additive manufacturing process. The authors utilized in-situ synchrotron x-ray imaging to observe keyhole dynamics and its influence on porosity formation. This is a unique monitoring technique because it provides a fundamental approach to understanding multiple variables that occur during the joining process. The authors did a good job of explaining studies that correspond to their research. In addition, due to the complex nature of materials joining, the authors managed many controlling factors efficiently and precisely by explaining how they occur in sequential order.

Overall, the article needs improvement on the organization or flow of the manuscript. It may be best to revert to the traditional technical manuscript layout of the following: Abstract, Introduction, Background/Literature Review, Experimental Method, Results/Discussion, and Conclusions. It would also be beneficial for the authors to address a key application that this research will target. There was a brief mention that this work has critical applications in aerospace, biomedical, automotive, and laser processing industries. However, there was no concrete description for how or where this research will be applied. This could be to the novelty of the study but there should be a clear goal to why this study is being conducted for the readers sake. For example, there was mention of improved material properties for the reduction of porosity in AM builds but there was no quantification of material properties in this study that could help fill the application gap for this manuscript. In addition, the authors of this paper may want to provide initial conditions for the material properties of the samples that they used for additional context for the reader. The LBPF is a complex joining process with many variables (as stated before), therefore it would be a good practice to show the actual chemical composition and mechanical properties of the substrates they used.

This paper is publishable with the revisions I have recommended and will support the reader's ability to follow the author's well-research thesis throughout the article by establishing/explaining to the reader a clear end application for this research.

Reviewer #3 (Remarks to the Author):

This work investigates the dynamics of keyhole porosity formation during laser powder bed fusion of an Al7A77 alloy using synchrotron X-ray imaging. I recommend considering the following points:

“...to reveal and elucidate different keyhole pore generation mechanisms and their corresponding keyhole melting regimes in LPBF.” --- It is not clear which is the novelty i.e. the important advance of these investigations with respect to analogous approaches previously published in the field: e.g.

<https://doi.org/10.1038/s41467-018-03734-7>, or DOI: 10.1126/science.aav4687.

“The typical processing-structure-property linkage for LPBF is: ...microstructural and mechanical property anisotropy.” --- these different drawbacks are dependent on the alloy types used and therefore must be focused on the type of alloys employed. The statement is excessively generalist. Moreover, the referred problems are not a consequence of the aspect studied in this work, namely the keyhole formation.

“Keyhole pores remaining in the final part may act as stress concentrators and sites for crack initiation and growth, making them potentially detrimental to fatigue life and other final component mechanical properties” --- Though considering porosity is important, hot isostatic pressing and post thermal treatment is usually required to minimize residual stresses and porosity, as well as to optimize the microstructure. Moreover, depending on the alloy and microstructure, keyhole pores may not always be detrimental for the mechanical properties. Their morphology (e.g. spherical or crack-like) may be decisive in such cases.

The setup employed opens up interesting studies about the dynamics of keyhole porosity formation. However, this work investigates this evolution during one track i.e. for one single LPBF layer. Thus, the direct transfer of the insights obtained to the LPBF process is questionable since the material obtained by LPBF is a consequence of a layer-by-layer manufacturing which involves re-heating of subsequently build layers (not considered here); this altering the porosity and the microstructure of the former layers.

“but it remains unclear how these physics extend to the LPBF process” --- to specify which specific variables are referred here and how may they be controlled during LPBF processing.

“Keyhole morphology variations across the (I) quasi-stable, (II) transition and (III) unstable keyhole regimes under different laser velocities...” --- it is unclear how are these stages defined and if these are concepts quantitatively distinguishable.

Ref. No.: NCOMMS-21-25887

Title: Keyhole fluctuation and pore formation mechanisms during laser powder bed fusion additive manufacturing

Dear Editor and Reviewers:

The authors appreciate the time and effort that you and the reviewers dedicated to providing feedback on the manuscript. The authors gratefully appreciate for all comments and suggestions, which helped improve the quality of the manuscript. All revisions in the manuscript are shown with track changes and revisions in Supplementary Information are addressed below. A point-by-point response to the reviewers' comments and suggestions are noted below in blue and purple.

Reviewer 1:

Keyhole fluctuation and porosity is an important question in additive manufacturing. This study provides more detailed and deeper insights into the bubble evolution before forming keyhole pores, and some conclusions are generally applicable to different materials, which is a significant step forward. While the manuscript is worthy of publication, the following comments are for the authors' consideration to further improve the manuscript:

The authors appreciate the reviewer's positive comments and constructive suggestions for revision.

1. In abstract, specifying the radial keyhole fluctuations to be ~10 kHz is misleading, because the frequency is 2.5 – 10 kHz in the results.

Author response: Thank you for pointing this out. The authors have clarified the frequency range in abstract as follows:

[Lines 18 – 20]: *“The findings support the hypotheses that: (i) keyhole porosity can initiate not only in unstable, but also transition keyhole regimes, created by high laser power-velocity conditions, causing fast radial keyhole fluctuations (2.5 – 10 kHz).”*

2. In Supplementary Line 93, the variable u should be v_l (laser scan speed).

Author response: Thank you for pointing this out. The authors have corrected the variable in **Supplementary Method** Line 93 (new line number is 113) as,

[Supplementary Line 113]: $L_{th}^* = \sqrt{\alpha / (v_l \cdot r_l)}$

3. In Supplementary Line 104, v_l may not be proper to be used in the Péclet number equation, and instead it should be the local flow velocity, which is usually higher than the laser scan speed but approximately at the same order though, because the physical meanings are different. Thus, it is not necessary to relate the mode of thermal transfer with the keyhole morphology in Line 124 of the manuscript.

Author response: Thank you for pointing this out. The authors agree with the reviewer's suggestion that the flow speed in the Péclet number equation should be the local flow speed rather than the laser scan speed. As the reviewer stated, these may be of similar magnitude but pose different physical meanings, and hence the authors removed discussion of the Péclet number (thermal transfer mode correlation in section *Keyhole collapse mechanism and related regime transitions* (Lines 128 – 129, 137 – 141) and the related derivations in **Supplementary Method** (Lines 103 – 105)).

4. It is interesting to see the good agreement between the theoretical inverse tangent relationship and the experimental results. There are several questions about this.

4-i) Can the normalized enthalpy product distinguish whether there is a keyhole for different materials?

Author response: The normalised enthalpy product can be used to identify the keyhole threshold for different materials. According to Fabbro, *J. Mater. Process. Technol.* 2019 and Cunningham *et al. Science.* 2019, the keyhole melting threshold can be defined using the front wall angle θ (keyhole mode when $\theta > 45^\circ$). Using the inverse tangent relationship derived in Fig.1b (extended to $\theta = 26.31^\circ$ as shown below), $\theta = \arctan[a \cdot (\Delta H/h_m \cdot L_{th}^* + b)]$, which includes data for Ti-6Al-4V from Cunningham *et al.* well below and around $\theta = 45^\circ$, the normalised enthalpy product value corresponding to $\theta = 45^\circ$ can be calculated as $\Delta H/h_m \cdot L_{th}^* = 3.6 \pm 0.8$ (Note, error was propagated from fitting parameters a and b , as given in Question4-iii). This keyhole threshold is consistent with the reports by Ye *et al. Adv. Eng. Mater.* 2019, where they found that the keyhole mode starts at $\Delta H/h_m \cdot L_{th}^* > 2$ based on melt pool depth data from SS316L, Inconel 625 and Ti-6Al-4V.

Fig.1b. Front keyhole wall (FKW) angle as a function of normalised enthalpy product for 9 datasets with 4 different materials.

References

1. Fabbro, Remy. Scaling laws for the laser welding process in keyhole mode. *Journal of Materials Processing Technology* 264 (2019): 346-351. doi: 10.1016/j.jmatprotec.2018.09.027.
2. Cunningham, Ross, et al. Keyhole threshold and morphology in laser melting revealed by ultrahigh-speed x-ray imaging. *Science* 363.6429 (2019): 849-852. doi: 10.1126/science.aav4687
3. Ye, Jianchao, et al. Energy coupling mechanisms and scaling behavior associated with laser powder bed fusion additive manufacturing. *Advanced Engineering Materials* 21.7 (2019): 1900185. doi:10.1002/adem.201900185.

4-ii) As we know that the ambient pressure influences the keyhole, is it possible to further consider this factor in the derivation?

Author response: This is an excellent suggestion. We have considered the pressure term (prior modelling done by Bidare *et al.* 2018a, b and recent work from Wang *et al.* 2021), however, we omitted this term as we maintain the gas/chamber pressure of +10 kPa (**Methods**, Line 408) during our LPBF experiments. Moreover, the *in situ* keyhole morphology data that we cited from Zhao *et al.* 2020, Cunningham *et al.* 2019, Hojjatzadeh *et al.* 2020 and Martin *et al.* 2019 was also collected under a constant atmospheric pressure (~ 1 atm). We plan to repeat the experiments with different atmospheric pressures on future beamtimes and further probe the pressure effect as the reviewer suggests, but this is out of the scope of this manuscript.

References

1. Bidare, Prveen, et al. Laser powder bed fusion at sub-atmospheric pressures. *International Journal of Machine Tools and Manufacture* 130 (2018). doi:65-72. 10.1016/j.ijmachtools.2018.03.007.

2. Bidare, Prveen, et al. Laser powder bed fusion in high-pressure atmospheres. The International Journal of Advanced Manufacturing Technology 99.1 (2018): 543-555. doi: 10.1007/s00170-018-2495-7.
3. Wang, Lu, et al. Mechanism of keyhole pore formation in metal additive manufacturing. arXiv preprint arXiv:2109.09480 (2021).
4. Zhao, Cang, et al. Critical instability at moving keyhole tip generates porosity in laser melting. Science 370.6520 (2020): 1080-1086. doi: 10.1126/science.abd1587
5. Cunningham, R. et al. Keyhole threshold and morphology in laser melting revealed by ultrahigh-speed x-ray imaging. Science. 363, 849–852 (2019). doi: 10.1126/science.aav4687.
6. Hojjatzadeh, S. Mohammad H., et al. Direct observation of pore formation mechanisms during LPBF additive manufacturing process and high energy density laser welding. International Journal of Machine Tools and Manufacture 153 (2020): 103555. doi:10.1016/j.ijmachtools.2020.103555
7. Martin, Aiden A., et al. Ultrafast dynamics of laser-metal interactions in additive manufacturing alloys captured by in situ X-ray imaging. Materials Today Advances 1 (2019): 100002. doi: 10.1016/j.mtadv.2019.01.001.

4-iii) I might miss it: what are the values of a, b and c? While the physical meaning of a seems obvious, what are the physical meanings of b and c?

Author response: The authors appreciate the comments. In terms of physical meanings, a is a constant coefficient ($a = K_3$), which is derived in **Supplementary Method** (Line 120). The second coefficient parameter, b , represents the threshold value in the normalised enthalpy product for the formation of a vapour depression. The constant, c , represents the difference between the actual and fitted front keyhole wall angle, which is nearly zero and can be removed without significantly affecting the fitting.

To make this clearer in the manuscript, we have removed the third fitting parameter, c , and added the values of a and b in the caption of Fig. 1 (Fig. 1 was also updated and is shown in above Question 4-i) as:

[Lines 117 – 119]: “*Curve fit is $\theta = \arctan[a \cdot (\Delta H/h_m \cdot L_{th}^* + b)]$ ($a = 0.29^{+0.04}_{-0.04}$, $b = -0.2^{+0.2}_{-0.6}$ with 95% confidence bounds), performed in Matlab using the Levenberg-Marquardt/least absolute residuals robust fitting algorithm”*

5. In Line 133, "the larger threshold for Al7A77" should be "smaller"? Because the value is ~8 for Al7A77 vs ~20 for TC4.

Author response: Thank you for pointing this out. The authors have corrected the sentences in the manuscript as (new line number is 147):

[Lines 147 – 151]: “*For Ti-6Al-4V^{16,23} and Al7A77 (Fig.1b, Fig.2d), we found this transition occurs at $\Delta H/h_m \cdot L_{th}^* \sim (8 \pm 3)$ or $\sim 60^\circ$ FKW angle, and $\Delta H/h_m \cdot L_{th}^* \sim (20 \pm 3)$ or $\sim 80^\circ$ front-wall angle, respectively. The larger threshold for Al7A77 is likely a combined result of its low absorptivity at ambient temperature (~ 0.15 vs. ~ 0.45), larger Brewster angle ($\sim 85^\circ$ ⁴⁴ vs. $\sim 80^\circ$, Supp. Fig S.5), and lower melting enthalpy ($h_m = 2.63 \text{ J} \cdot \text{mm}^{-3}$ vs. $6.26 \text{ J} \cdot \text{mm}^{-3}$).*”

6. For the keyhole fluctuation period and frequency in Fig.2, would it be better to perform Fourier analysis to extract the significant patterns? As there are multiple forces acting on the keyhole, they may have different patterns and thus induce different fluctuation frequencies.

Author response: The authors appreciate this inspiring comment from the reviewer, and we agree that it is very likely that there are multiple forces acting on the keyhole and using Fourier analysis might resolve this. In our initial study, we performed Fourier analysis on our synchrotron X-ray data (e.g., keyhole depth, width), but no conclusive answer due to the limitation in the field of view (FOV). Although of concern for peak finding as well, aliasing would also likely corrupt the results of a Fourier analysis (the authors’ Nyquist frequency being 25 kHz, as stated in Line 453). This might be improved in future work by having the FOV track the laser, imaging using a Lagrangian frame of reference.

7. In Fig.2b, it may be improper to claim the shaded blue as the transition regime for all the four alloys there, because the authors claimed the threshold for the stable/unstable keyhole transition can vary significantly between alloys.

Author response: Thank you for pointing this out. The authors agree that the shaded blue region could suggest that the transition regime is the same for all four materials, therefore, we have removed the shaded blue region in Fig 2b as shown below (also removed the description in Fig.2b caption):

Fig. 2b. Average period between successive peaks/valleys in keyhole width as a function of normalised enthalpy product.

8. It is good to see the good agreement between the model and experiments on the bubble size evolution. However, hydrogen diffusion still has no direct experimental evidence, and the model has not been validated in similar scenarios. Better to achieve some experimental evidence (maybe atom probe tomography can detect the hydrogen on the keyhole pore surfaces) or use some models that have been validated in additive manufacturing scenarios.

Author response: The authors appreciate this comment from the reviewer and agree that it is an unproven hypothesis that hydrogen diffusion plays a key role, although it is based on the authors' decades of experience in partially analogous welding and casting processes. We did try to think of various methods of trying to quantify the hydrogen content (e.g., MS (mass spectrometry) to APT (atom probe tomography)) but none of the methods was viable.

To clarify, we have altered the text to make it very clear that this is a hypothesised mechanism that might help explain the pore growth behaviour observed in our X-ray results, *i.e.* the reduction in the rate of shrinkage and subsequent bubble size stabilization. To back up our hypothesis, we also added the following citations (Lines 287 – 292) from Matsunawa *et al.* 2003 who measured ~ 3 – 12% hydrogen content in porosity with keyhole in aluminium alloy welding; Atwood *et al.* 2000, who built and verified a diffusion-controlled hydrogen pore model; and Lee and Hunt 1997, 2001, who observed and modelled hydrogen diffusion during pore formation in directional solidification of aluminium.

In addition, hydrogen has been found to be present in the virgin substrate, powder particles, as well as the environmental chamber (Anderson *et al.* 2018) in LPBF. During cooling, the melt at the advancing solidification front can become supersaturated with hydrogen (hydrogen solubility in liquid aluminium at the melting temperature is approximately 10 times larger than that in solid aluminium, reported by Christian *et al.* 2015), driving hydrogen diffusion from the melt into the bubble with a high diffusion rate (mass diffusion rates are $D_h(T_l) = 1.0943 \times 10^{-7} m^2 \cdot s^{-1}$ and $D_h(T_v) = 1.1302 \times 10^{-5} m^2 \cdot s^{-1}$ (Anyalebechi. 2003)).

References

1. Matsunawa, A., et al. Porosity formation mechanism and its prevention in laser welding. *Welding international* 17.6 (2003): 431-437. doi:10.1533/wint.2003.3138.
2. Atwood, R. C., et al. "Diffusion-controlled growth of hydrogen pores in aluminium–silicon castings: in situ observation and modelling." *Acta materialia* 48.2 (2000): 405-417. doi: 10.1016/S1359-6454(99)00363-8.
3. Lee, P. D., and J. D. Hunt. Hydrogen porosity in directional solidified aluminium-copper alloys: in situ observation. *Acta materialia* 45.10 (1997): 4155-4169. doi: 10.1016/S1359-6454(97)00081-5.
4. Lee, P. D., and J. D. Hunt. Hydrogen porosity in directionally solidified aluminium–copper alloys: a mathematical model. *Acta materialia* 49.8 (2001): 1383-1398. doi:10.1016/S1359-6454(01)00043-X.
5. Anderson, Iver E., Emma MH White, and Ryan Dehoff. Feedstock powder processing research needs for additive manufacturing development. *Current Opinion in Solid State and Materials Science* 22.1 (2018): 8-15. doi: 10.1016/j.cossms.2018.01.002.
6. Anyalebechi, P. N. Critical review of reported values of hydrogen diffusion in solid and liquid aluminum and its alloys. TMS, 2003.
7. Weingarten, Christian, et al. Formation and reduction of hydrogen porosity during selective laser melting of AlSi10Mg. *Journal of Materials Processing Technology* 221 (2015): 112-120. doi:10.1016/j.jmatprotec.2015.02.013.

Reviewer 2:

This article provides a detailed study of how porosity develops in the laser powder-bed fusion additive manufacturing process. The authors utilized in-situ synchrotron x-ray imaging to observe keyhole dynamics and its influence on porosity formation. This is a unique monitoring technique because it provides a fundamental approach to understanding multiple variables that occur during the joining process. The authors did a good job of explaining studies that correspond to their research. In addition, due to the complex nature of materials joining, the authors managed many controlling factors efficiently and precisely by explaining how they occur in sequential order.

Overall, the article needs improvement on the organization or flow of the manuscript. It may be best to revert to the traditional technical manuscript layout of the following: Abstract, Introduction, Background/Literature Review, Experimental Method, Results/Discussion, and Conclusions. It would also be beneficial for the authors to address a key application that this research will target. There was a brief mention that this work has critical applications in aerospace, biomedical, automotive, and laser processing industries.

However, there was no concrete description for how or where this research will be applied. This could be to the novelty of the study but there should be a clear goal to why this study is being conducted for the readers sake. For example, there was mention of improved material properties for the reduction of porosity in AM builds but there was no quantification of material properties in this study that could help fill the application gap for this manuscript. In addition, the authors of this paper may want to provide initial conditions for the material properties of the samples that they used for additional context for the reader. The LBPF is a complex joining process with many variables (as stated before), therefore it would be a good practice to show the actual chemical composition and mechanical properties of the substrates they used.

This paper is publishable with the revisions I have recommended and will support the reader's ability to follow the author's well-research thesis throughout the article by establishing/explaining to the reader a clear end application for this research.

Author response: The authors appreciate the comments and constructive suggestions for revision, and we've responded below with points 1 to 3.

1. Overall, the article needs improvement on the organization or flow of the manuscript. It may be best to revert to the traditional technical manuscript layout of the following: Abstract, Introduction, Background/Literature Review, Experimental Method, Results/Discussion, and Conclusions.

Author response: The authors appreciate the comment and agree that the layout is not the traditional one, but we are following the style requirements of Nature Communications.

2. It would also be beneficial for the authors to address a key application that this research will target. There was a brief mention that this work has critical applications in aerospace, biomedical, automotive, and laser processing industries. However, there was no concrete description for how or where this research will be applied. This could be to the novelty of the study but there should be a clear goal to why this study is being conducted for the readers sake. For example, there was mention of improved material properties for the reduction of porosity in AM builds but there was no quantification of material properties in this study that could help fill the application gap for this manuscript.

Author response: The authors appreciate this comment, and the authors made some statements on the applications of this study in Lines 365 – 370 of manuscript as shown below in purple:

[Lines 365 – 370]: *“Our findings on keyhole fluctuation and bubble dynamics provide critical guidance (e.g., bubble growth/shrinkage rate, pore location and size) to achieve in situ pore elimination by remelting^{25,61} upon the dual-laser LPBF machines⁶² or hybrid LPBF⁶³, and pore suppression via real-time control of keyhole dynamics (e.g., beam oscillation⁶⁴) in a broad range of high-energy-beam processing techniques (e.g., electron beam melting⁶⁵, keyhole laser welding⁶⁴ and laser drilling⁶⁶).”*

We agree that in the future performing large scale builds and subsequent mechanical testing to carefully quantify the impact of these insights on properties is a laudable next step, but outside the scope of this study. The focus of our paper is to improve understanding of keyhole behaviour to minimise pore formation and hence improve the mechanical properties. The correlation between porosity and mechanical properties, such as ductility/elastic modulus (Martin *et al.* Nature. 2017), as well as fatigue (Tammis-Williams *et al.* 2017) are well studied, and were also added in the paper (**Introduction**, Lines 49 – 50).

References

1. Tammis-Williams, S., et al. The influence of porosity on fatigue crack initiation in additively manufactured titanium components. Scientific reports 7.1 (2017): 1-13. doi: 10.1038/s41598-017-06504-5.
2. Martin, John H., et al. 3D printing of high-strength aluminium alloys. Nature 549.7672 (2017): 365-369. doi: 10.1038/nature23894.

3. In addition, the authors of this paper may want to provide initial conditions for the material properties of the samples that they used for additional context for the reader. The LBPF is a complex joining process with many variables (as stated before), therefore it would be a good practice to show the actual chemical composition and mechanical properties of the substrates they used.

Author response: The authors appreciate this comment and added a table (**Supplementary Table 2**) with the chemical composition for the 7A77 powders, shown below and now referenced on Supplementary Lines 188 – 189:

Supplementary Table 2. Material composition of the Al7A77 powder²⁸.

Aluminium	Chromium	Copper	Magnesium	Zinc	Zirconium	Trace Elements
Balance	< 0.1 %	1.1 - 2.1%	1.8 - 2.9%	4.5 - 6.1%	0.5 - 2.8%	< 0.15%

As the 7A77 powder and pure aluminium substrate are commercially supplied and are standard alloys, the authors did not measure the mechanical properties; These values are available in the literature, e.g. in ASTM handbook *Volume 2B: Properties and Selection of Aluminium Alloys* and the product company data sheet (HRL Laboratories. *Aluminium 7A77 Data Sheet*). In the manuscript, the authors had listed some material properties in the Section of Methods, Lines 412 – 417 (shown below in italic).

[Lines 412 – 417]: “*The commercially Al7A77 powder (HRL Laboratory, USA, material composition shown in Supplementary Table 2) with a particle size range of 15 – 45µm, and pure aluminium (Goodfellow, UK) plate with purity of 99.99% that sandwiched between two 1 mm thickness glassy carbon plates (HTW, Germany), were used in this study with the process parameters shown in Table 1. Thermophysical properties of the materials are shown in Supplementary Table 3.*”

Reference

1. HRL Laboratories. Aluminium 7A77 Data Sheet. https://www.hrl.com/products-services/materials/_assets/7A77-data-sheet.pdf.

Reviewer 3:

This work investigates the dynamics of keyhole porosity formation during laser powder bed fusion of an Al7A77 alloy using synchrotron X-ray imaging. I recommend considering the following points:

Author response: The authors appreciate the comments and constructive suggestions for revision.

1. “...to reveal and elucidate different keyhole pore generation mechanisms and their corresponding keyhole melting regimes in LPBF.” --- It is not clear which is the novelty i.e. the important advance of these investigations with respect to analogous approaches previously published in the field: e.g. <https://doi.org/10.1038/s41467-018-03734-7>, or DOI: 10.1126/science.aav4687.

Author response: Thanks for reviewer’s feedback on this sentence. To clarify the novelty of our work, we modified this sentence in manuscript (Lines 362 – 365), please see purple text below.

[Lines 362 – 365]: “*In summary, this manuscript reveals the lifetime dynamics of keyhole pore (growth, shrinkage, migration, interaction with microstructure and capture by advancing solidification front), introducing a threshold, the normalised enthalpy product, to reveal and elucidate different keyhole pore generation mechanisms and their corresponding keyhole melting regimes under stable, transition and unstable conditions in LPBF.*”

For the two references, Leung *et al.* 2018 focused on pore formation under overhang conditions, Cunningham *et al.* 2019 focused on the transition from conduction to keyhole melting (onset of keyhole melting). Here, we step further into three different keyhole melting modes (stable, transition and unstable, shown in Fig.1a), keyhole fluctuation dynamics, and bubble evolution with gas diffusion mechanism before forming keyhole porosity.

We also hope the reviewer agree that our work has provided 5 new insights into the keyhole pore formation mechanisms during LPBF, see lines 371 – 398 in **Discussion**.

References

1. Leung, C. L. A. et al. In situ X-ray imaging of defect and molten pool dynamics in laser additive manufacturing. *Nat. Commun.* 9, (2018). doi: 10.1038/s41467-018-03734-7.
2. Cunningham, R. et al. Keyhole threshold and morphology in laser melting revealed by ultrahigh-speed x-ray imaging. *Science.* 363, 849–852 (2019). doi: 10.1126/science.aav4687.

2. “The typical processing-structure-property linkage for LPBF is: ...microstructural and mechanical property anisotropy.” --- these different drawbacks are dependent on the alloy types used and therefore must be focused on the type of alloys employed. The statement is excessively generalist. Moreover, the

referred problems are not a consequence of the aspect studied in this work, namely the keyhole formation.

Author response: The authors appreciate this comment, and have altered the sentence to clarify that these are typical behaviours and depend upon alloy as the reviewer states:

[Lines 32 – 34]: “*The typical processing-structure-property linkage for LPBF is: steep thermal gradients and high cooling rates³ ($\sim 10^4 - 10^6 \text{ K}\cdot\text{s}^{-1}$) favouring fine, columnar grains oriented along the build direction, producing as-printed LPBF parts that **typically** exhibit increased strength, reduced ductility, and increased microstructural and mechanical property anisotropy⁴, **depending on the alloy systems.**”*

The major findings of this paper (e.g., keyhole regime transitions, keyhole fluctuation and collapse) are also applicable to a range of different materials, as illustrated in Fig 1b, and Fig 2b, c for Ti-6Al-4V, Inconel 718 and Stainless steel 304, as the first paragraph of **Introduction**, the content was not limited to the specific alloy used in the current experiments (A17A77). Also, a brief introduction to the typical processing-structure-property linkage in LPBF, as established by DebRoy *et al.* 2018, Thijs *et al.* 2010, Amato *et al.* 2012, Cauwenbergh *et al.* 2021, might aid reader understanding (Nat. Commun. is a multidisciplinary journal), and lead into the more detailed focus of this research, keyhole dynamics and porosity in the second paragraph of **Introduction** (Lines 35 – 50).

References:

1. DeBroy, T. *et al.* Additive manufacturing of metallic components – Process, structure and properties. Prog. Mat. Sci. 92, (2018). doi: 10.1016/j.pmatsci.2017.10.001.
2. Thijs, L. *et al.* A study of the microstructural evolution during selective laser melting of Ti-6Al-4V. Acta Materialia. 58, (2010). doi: 10.1016/j.actamat.2010.02.004.
3. Amato, *et al.* Microstructures and mechanical behaviour of Inconel 718 fabricated by selective laser melting. Acta Materialia. 60, (2012). doi: 10.1016/j.actamat.2011.12.032.
4. Cauwenbergh, P. Van *et al.* Unravelling the multi-scale structure-property relationship of laser powder bed fusion processed and heat treated AlSi10Mg. Scientific Reports. 11, (2021). doi: doi:10.1038/s41598-021-85047-2.

3. “Keyhole pores remaining in the final part may act as stress concentrators and sites for crack initiation and growth, making them potentially detrimental to fatigue life and other final component mechanical properties” --- Though considering porosity is important, hot isostatic pressing and post thermal treatment is usually required to minimize residual stresses and porosity, as well as to optimize the microstructure. Moreover, depending on the alloy and microstructure, keyhole pores may not always be detrimental for the mechanical properties. Their morphology (e.g. spherical or crack-like) may be decisive in such cases.

Author response: The authors appreciate this comment and agree with the reviewer that post processing treatments (e.g., hot isostatic processing, HIPing) are important for additive manufacturing components to close porosity. However, the post processing may increase production time and cost, and prior research has found that the argon containing gas pores (argon is widely used as shielding gas in LPBF and is the major content of keyhole pores) can reappear and grow after HIPing during high temperature treatments (Tamas-Williams *et al.* 2016).

The authors also agree with the reviewer that the keyhole pores may not always be detrimental with regarding to some specific applications. Prior work reported that the porosity deteriorates parts’ mechanical properties such as fatigue crack (Tamas-Williams, S., *et al.* 2017), and ductility and elastic modulus (Martin *et al.* 2017). It is therefore, we have used “may” and “potentially” in the text to indicate this:

[Lines 47 – 50]: “*Keyhole pores remaining in the final part may act as stress concentrators and sites for crack initiation and growth, making them potentially detrimental to fatigue life¹⁷ and other final component mechanical properties^{18,19}.*”

References

1. Tammas-Williams, Samuel, et al. Porosity regrowth during heat treatment of hot isostatically pressed additively manufactured titanium components. *Scripta Materialia* 122 (2016): 72-76. doi: 10.1016/j.scriptamat.2016.05.002
2. Tammas-Williams, S., et al. The influence of porosity on fatigue crack initiation in additively manufactured titanium components. *Scientific reports* 7.1 (2017): 1-13. doi:10.1038/s41598-017-06504-5

4. The setup employed opens up interesting studies about the dynamics of keyhole porosity formation. However, this work investigates this evolution during one track i.e. for one single LPBF layer. Thus, the direct transfer of the insights obtained to the LPBF process is questionable since the material obtained by LPBF is a consequence of a layer-by-layer manufacturing which involves reheating of subsequently build layers (not considered here); this altering the porosity and the microstructure of the former layers.

Author response: The authors appreciate this comment and the authors agree with the reviewer that the single-track experiments do not cover the reheating effect during the multi-layer/track scanning of LPBF. Unfortunately, performing high speed radiography imaging of multi-layer/track builds and capturing quickly enough to cover the effect of heat build-up is not currently possible for several reasons, the main two being: i) Off-loading camera data after imaging a track can take up to 10 minutes for typical frame rates/durations, which is considerably longer than the time between laser passes/tracks in an LPBF build and eliminates the effect of residual heating; ii) Image quality for subsequent layers is much worse due to sintered powder on the front/back of the prior built track, preventing automated quantification. We have now built a second version of *in situ* LPBF replicator rig that allows multiple layers and hope to address the reviewer's excellent suggestion in future studies if we can overcome these obstacles.

In addition, as shown by recent publications (Zhao *et al. Science*. 2020, Khairallah *et al. Science*. 2020, Cunningham *et al. Science*. 2019, Martin *et al. Nat. Commun*. 2019, Gan *et al. Nat. Commun*. 2021, Sun *Jom*. 2020, Hojjatzadeh *et al. Int. J. Mach. Tools Manuf*. 2020, Kouraytem *et al. Phys. Rev. Appl*. 2019), the single-track experiments provide the same mechanistic insights on keyhole fluctuation, keyhole collapse and keyhole induced porosity formation with the multi-layer scanning.

References

1. Zhao, Cang, et al. Critical instability at moving keyhole tip generates porosity in laser melting. *Science* 370.6520 (2020): 1080-1086. doi: 10.1126/science.abd1587
2. Khairallah, Saad A., et al. Controlling interdependent meso-nanosecond dynamics and defect generation in metal 3D printing. *Science* 368.6491 (2020): 660-665. doi:10.1126/science.aay7830
3. Cunningham, R. et al. Keyhole threshold and morphology in laser melting revealed by ultrahigh-speed x-ray imaging. *Science*. 363, 849–852 (2019). doi: 10.1126/science.aav4687.
4. Martin, Aiden A., et al. Dynamics of pore formation during laser powder bed fusion additive manufacturing. *Nature communications* 10.1 (2019): 1-10. doi: 10.1038/s41467-019-10009-2.
5. Gan, Zhengtao, et al. Universal scaling laws of keyhole stability and porosity in 3D printing of metals. *Nature communications* 12.1 (2021): 1-8. doi:10.1038/s41467-021-22704-0
6. Sun, Tao. Probing Ultrafast Dynamics in Laser Powder Bed Fusion Using High-Speed X-Ray Imaging: A Review of Research at the Advanced Photon Source. *JOM* 72.3 (2020): 999-1008. doi: 10.1007/s11837-020-04015-9
7. Hojjatzadeh, S. Mohammad H., et al. Direct observation of pore formation mechanisms during LPBF additive manufacturing process and high energy density laser welding. *International Journal of Machine Tools and Manufacture* 153 (2020): 103555. doi:10.1016/j.ijmachtools.2020.103555
8. Kouraytem, Nadia, et al. Effect of laser-matter interaction on molten pool flow and keyhole dynamics. *Physical Review Applied* 11.6 (2019): 064054. doi:10.1103/PhysRevApplied.11.064054

5. “but it remains unclear how these physics extend to the LPBF process” --- to specify which specific variables are referred here and how may they be controlled during LPBF processing.

Author response: The authors appreciate this comment and have rewritten the sentence to clarify:

[Lines 65 – 67]: “For the latter, previous studies^{30,31} explored the influence of evaporation and condensation on the dynamics of water-vapour bubbles in a superheated liquid, and effect of dissolved gas diffusion on bubble growth in casting^{32,33}, but it remains unclear how **evaporation, vapour condensation, and dissolved gas diffusion affect bubble evolution in LPBF.**”

In terms of how evaporation, vapour condensation, and/or dissolved gas diffusion can be controlled to mitigate keyhole porosity, the authors considered mentioning two potential approaches: i) Degassing the packed powder and/or chamber to reduce hydrogen sources during LPBF; ii) Controlling the atmospheric pressure to tune the keyhole in steady or transition modes, and further adjust the keyhole pressure, temperature and content (i.e., less inert gas molecules at sub-atmospheric pressures (Bidare *et al.*, 2018)) to govern bubble evolution.

References

1. Bidare, Prveen, et al. Laser powder bed fusion at sub-atmospheric pressures. *International Journal of Machine Tools and Manufacture* 130 (2018). doi:65-72. 10.1016/j.ijmachtools.2018.03.007.

6. “Keyhole morphology variations across the (I) quasi-stable, (II) transition and (III) unstable keyhole regimes under different laser velocities...” --- it is unclear how are these stages defined and if these are concepts quantitatively distinguishable.

Author response: The authors appreciate this comment and have added the keyhole morphology change in the sentence to make it clear as below:

[Line 115]: “Fig.1. Keyhole collapse mechanism and related keyhole melting regime transitions in LPBF. **a Keyhole morphology variations from wide and shallow to narrow and deep across the (I) quasi-stable, (II) transition and (III) unstable keyhole regimes under different laser velocities.**”

The authors also defined changes in morphology and pore formation across the keyhole melting regimes in Lines 92 – 98 and Lines 152 – 156, respectively:

[Line 92 – 98]: “We observed that the keyholes change in morphology from wide and shallow to narrow and deep (Fig.1a, Supplementary Fig. 2a and Supplementary Movies 1 – 12). Simultaneously, bubbles first form at the RKW, then prevail at the bottom of keyhole once the keyhole becomes deep and narrow (Fig.1a, Supplementary Fig. 2a). Those findings indicate that the transition from a stable to unstable keyhole melting may be more nuanced than previously suggested^{16,25} (discussed in detail later).”

[Line 152 – 156]: “In addition, we find that there can be an extended transition regime (II) between the stable (I) and unstable (III) keyhole regimes under high-power-velocity (high-PV). Pores begin to form in this transition regime, and initiate at the RKW rather than at the bottom of the keyhole (typical in III), which was also observed during laser welding of aluminium alloys¹⁵ and low carbon steel⁴⁵, as well as LPBF of Ti-6Al-4V⁴¹.”

Minor revisions from the authors:

All the revisions were tracked in manuscript and the associated explanations were added below.

1. The following changes were made to meet the formatting requirements of Nature Communications.
 - i) The final sentence of **Abstract** was changed to include a brief summary of the major results.;

[Line 24 – 25]: “*The keyhole fluctuation and bubble evolution mechanisms revealed here may guide the development of control systems for minimising keyhole porosity*”.

ii) The titles of Figs.1 – 4 were changed to bold fonts, panels that labelled using the a), b) and c) were changed to a, b, c, convention.

iii) The fonts of Figs.1 – 4 were changed to Arial, and the font size is set between 5pt and 8pt. To keep same font with the Figs, the fonts in manuscript were also changed to Arial.

iv) Supplementary items were cited as “Supplementary Fig, Supplementary Discussion, Supplementary Movie, and Supplementary Table” in the manuscript.

v) One subheading in Methods was changed to fit requirement of “Subheadings should be no longer than 60 characters”:

[Line 419 – 420]: “*Image and data processing pipeline*”.

2. Section of **Keyhole-induced bubble lifetime dynamics in LPBF**. Fig. 4a. The authors added one more frame of the radiography (time $t = 140 \mu\text{s}$) in Fig. 4a to keep same time span of Fig. 4b and corrected the time legend as 0 – 140 μs as shown below:

Fig.4a. Tracking and modelling of keyhole induced bubble dynamics. Colour map tracking for keyhole and bubble under low **a** and high **b** laser velocities, corresponding to regimes (III) and (II), respectively.

3. The authors corrected the typo of “vapor” to “vapour” in line 377. The authors corrected the typo of “leaning” to “learning” in the Lines 442, 443 and caption of Supplementary Fig. 14.

4. As per the new requirement of acknowledgment statement for publications from Advanced Photon Source (APS), the authors added the following sentence in the **Acknowledgement**:

“This research used resources of the Advanced Photon Source, a U.S. Department of Energy (DOE) Office of Science User Facility, operated for the DOE Office of Science by Argonne National Laboratory under Contract No. DE-AC02-06CH11357.”

5. Supplementary Fig. 12 (d-ii). The colour legend of grain size (area in μm^2) was added.

Supplementary Fig. 12d-ii Grain size spread shown with rainbow colour (ii).

6. *Supplementary Fig. 13d*. The three axes labels were changed from “x”, “y”, and “z” to “x”, “y”, and time “t”:

Supplementary Fig. 13d Frame stack integration with voxel thresholding in time (t) domain. Keyhole is marked by green boundary while pores are marked by red boundary.

7. Supplementary Table 3. The thermal diffusivity unit was corrected as $10^{-6} \text{ m}^2 \cdot \text{s}^{-1}$.

REVIEWERS' COMMENTS

Reviewer #1 (Remarks to the Author):

The authors have well addressed all my previous comments, and I am supportive to the acceptance and publication of this manuscript.

Reviewer #2 (Remarks to the Author):

The authors addressed all of my comments and have shown them in the completed manuscript. I am satisfied with their contributions to the field and addressing my comments.

Reviewer #3 (Remarks to the Author):

It is not clear which is the novelty i.e. the important advance of these investigations with respect to analogous approaches previously published in the field: e.g. <https://doi.org/10.1038/s41467-018-03734-7>, or DOI: 10.1126/science.aav4687.

The novelty of this work is not clearly clarified in the introduction with respect to prior analogous investigations. The advantages of the methodology employed have been previously reported. Papers published by the journal aim to represent important advances within a field which open up new research windows. This is not clear from the authors explanation (not included in the text) since it involves the study of pore formation under different situations but not a new concept:

Leung et al.2018 focused on pore formation under overhang conditions, Cunningham et al. 2019 focused on the transition from conduction to keyhole melting (onset of keyhole melting). Here, we step further into three different keyhole melting modes (stable, transition and unstable, shown in Fig.1a), keyhole fluctuation dynamics, and bubble evolution with gas diffusion mechanism before forming keyhole porosity.“

„keyhole morphology variations across the (I) quasi-stable, (II) transition and (III) unstable keyhole regimes under different laser velocities...”---it is unclear how are these stages defined and if these are concepts quantitatively distinguishable. Quantitative kinetics defining the onsets of these regimes are still required to support the conclusions obtained.